# Temporal and spatial features of the palace building space of Qinghai's Kumbum Monastery and its evolution

**Jing Zhang**[1]*, Yunying Ren[2], Xiaofan An[3]

**1** Department of Environmental Design, School of Art and Media, Xi'an Technological University, Xi'an, Shaanxi, China, **2** Department of Urban and Rural Planning, College of Architecture, Xi'an University of Architecture and Technology, Xi'an, Shaanxi, China, **3** Northwest Engineering Corporation Limited, Power China, Xi'an, Shaanxi, China

* 357601193@qq.com

**Data Availability Statement:** Figs 8, 10, 11 are availble from the Figshare database S1, S2, the URLs: https://figshare.com/articles/figure/S1_Figure_of_surveying_and_mapping_of_Golden_Tile_Palace_dwg_S2_Figure_of_surveying_and_

## Abstract

Kumbum Monastery is the centre of Tibetan Buddhism and religious culture in Qinghai. Its palace buildings are the typical examples of Tibetan Buddhist monastery buildings and living fossils of Tibetan social history. This study selected 12 palace buildings of Kumbum Monastery as the study objects, used typological approaches and numerical method to analyse their spatial features, classified these features into four types (the ring road surrounding Dugang-style palace space, single-sided eaves of the Duguang-style palace space, three-stage palace space with cloisters connecting to Dugang, and other variants) according to their spatial structures, and discussed the temporal evolution of the spatial features from spatial and temporal perspectives to obtain the development process for the palace buildings of Kumbum Monastery. The analysis showed that: 1) Because the Han people migrated to Qinghai, the architectural space of the monastery followed the practices of Tibet in the same period and began to adopt the practices of the Han people under the effect of religious indoctrination and sociopolitical influence of the Ming dynasty. 2) Due to the influence of religious development facilitated by the political environment in the Qing dynasty, all the palace buildings of Kumbum Monastery adopted the practice of monasteries in Lhasa. After being implicated by the political rebellion, the monastery initiated to add the spatial layout elements of the buildings in Qinghai. Therefore, the monastery was obtaining the cues from the Han culture. 3) The dominance of religious significance over the spatial designing of the palace buildings of Kumbum Monastery gradually shifted to the political dominance. This paper revealed the spatial and temporal features and evolution of the palace building space, explored the generation process of the palace buildings space of Tibetan Buddhist monasteries in Qinghai, and provided a reference for the static conservation and dynamic development of Tibetan Buddhist monastery buildings in Qinghai.

mapping_of_Great_Chanting_Hall/16823605 And the rest data are within the manuscript.

**Funding:** Yunying Ren received NSFC,the grant number: 52078404, the full name of the funder is 'National Nature Science Foundation of China'. URL:http://output.nsfc.gov.cn/fundingQuery The funders had no role in study design, data collection and analysis, decision to publish, or preparation of the manuscript.

**Competing interests:** The authors have declared that no competing interests exist.

# Introduction

## Background and aim

Kumbum Monastery is located in a narrow valley in Lushaer, Huanzhong County, Xining City, Qinghai Province, China. This geographical location represents the origin of the Later Period of Tibetan Buddhism [1–3]and is a typical area with a history of multicultural exchange due to the intersection of ancient traffic routes, such as the Silk Road, the ancient Tang-Tubo Road, and the Tuyuhun (a branch of ancient Mongolian) migration route in Amdo [4, 5] (Fig 1). As the centre of Tibetan Buddhism and religious culture in Qinghai, it occupies an area of 480.850m2 and has 52 palace buildings, 9,300 rooms, and a constructed area of 100,000 m². The architectures of Kumbum Monastery reflect the religious culture of Tibetan Buddhist and the regional culture of Qinghai. Among these buildings, the grand halls of Buddhas and the Buddhist colleges are used for the monastery's main spatial functions, such as Buddhist activities and religious teaching. These two buildings constitute the core construction type of the monastery, that is, the palace building [6–9]. The palace buildings are not only the representatives of Kumbum Monastery and Tibetan Buddhist architecture but also are the living fossils of the Tibetan traditional society and history of Qinghai. Kumbum Monastery was constructed over a period of 563 years (1379–1942). During the three periods of its construction (Period I: 1379–1648, Period II: 1649–1873, and Period III: 1874–1942), construction activities (construction, reconstruction, and renovation) of the palace buildings were always prioritized [10]. The construction of the palace buildings continued through the periods of Ming and Qing dynasties and the Republic of China. Therefore, their spatial features reflect the rules and formal patterns of the environmental and topographical conditions and human relations, which suggest both the localisation of Tibetan Buddhism in the Qinghai region and the socio-political and economic development, transportation, and ethnic features of different historical periods.

This paper focus on the 12 palace buildings in Kumbum Monastery through typological approach, and aims to clarify their spatial features from the viewpoint of the spatial elements and their combination. As well as explores their evolution process and their relation to regional and historical contexts.

## Literature review

In previous studies on architectures of Kumbum Monastery, the overall spatial features of the monastery were examined [11, 12]. Moreover, descriptive studies on the architectural structure, material characteristics, and decorative arts of a few major individual buildings have been conducted [13–15]. Considering the architectural aesthetic characteristics, Yang analysed the form of the monastery buildings through field surveys and immediate observations [16]. From 1992 to 1996, the Chinese government funded the large-scale mapping and restoration of Kumbum Monastery, and the obtained mapping data provided this paper with important information. Subsequently, Ran [6] and Chen [8], by using architectural history and typology, respectively, have classified the Tibetan Buddhist architectures by using the data from a study on Kumbum Monastery. In addition, Yang [16] have thoroughly analysed the architectural structures, material characteristics, and decorative arts of a part of the main single building of Kumbum Monastery. Zhang [17] analysed the spatial features of the palace buildings of Tibetan Buddhist monasteries and classified the buildings according to their architectural types. The historical origin of the palace buildings of Kumbum Monastery was discussed and analysed [18, 19]. The results of these studies [6–9, 18, 19] are the frame of reference for selecting the types of palace buildings in this study.

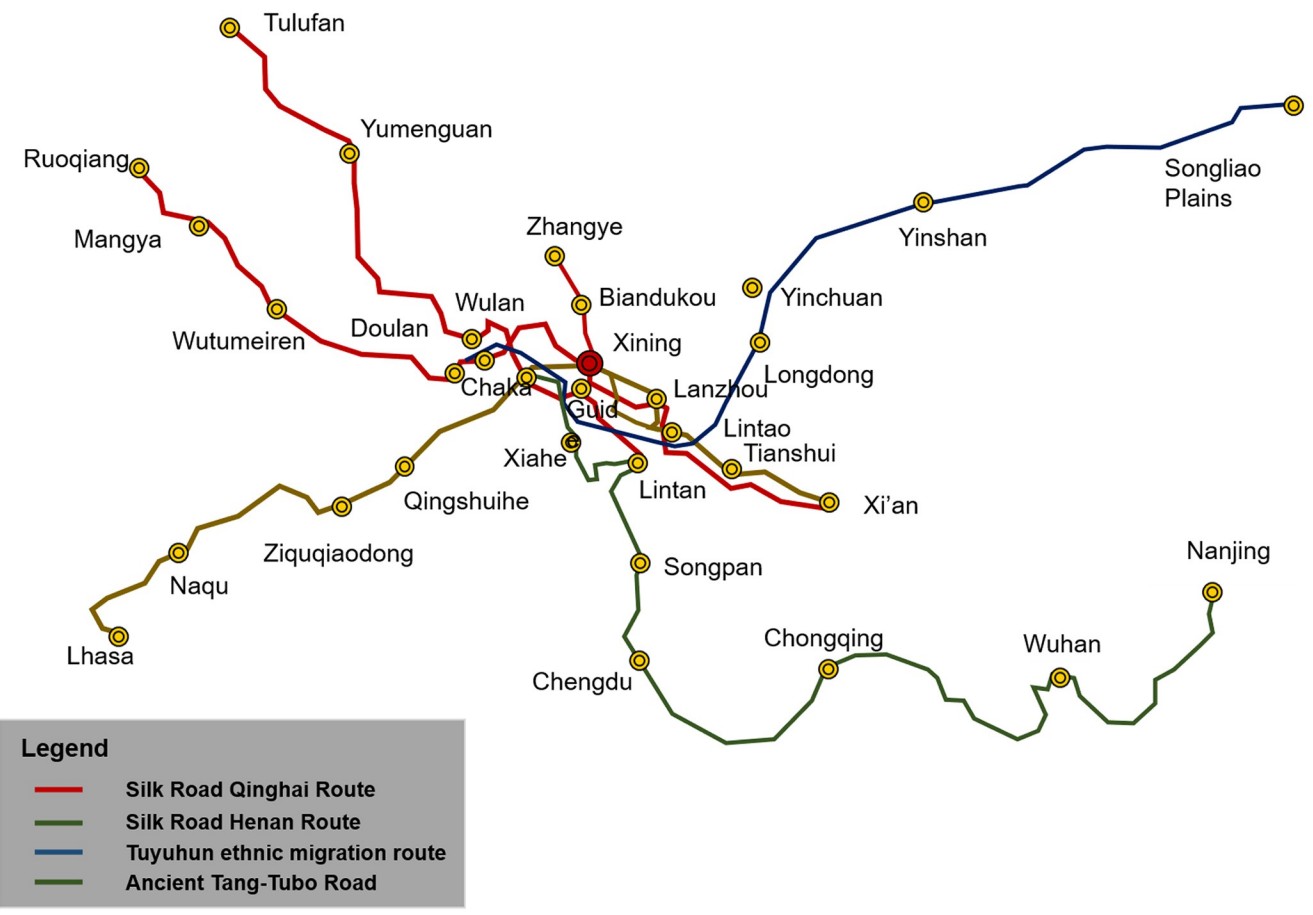

**Fig 1. Ancient traffic routes in Qinghai province (202 B.C.–907 A.D.).**

The aforementioned studies are crucial for understanding the development of the monastery and the morphological and technical characteristics of the monastery's buildings. However, few studies have discussed the spatial evolution of building types and evolution process of the monastery on the basis of data analysis. The exploration of the overall evolutionary law of the monastery's buildings is limited because of the insufficient attention given to the references written in Tibetan language, which is not conducive to a comprehensive understanding of their generation process. Additionally, in the face of the crisis of historic building conservation and dissipation of regional traditional culture with rapid urbanisation, the study and conservation of historic buildings with minority characteristics have become a concern for academicians. The existing 655 Tibetan Buddhist monastic buildings in Qinghai, which are an important part of the Amdo Region and the base for Tibetan Buddhism revival [20, 21], must be urgently and systematically investigated. Kumbum Monastery is a religious centre and typical representative of the monastic buildings of Qinghai.

With this shared background, this paper focus on 12 palace buildings in Kumbum Monastery, and explores their spatial characteristics by discussing the spatial elements (interior ring road), combination of spatial elements (Buddha halls, chanting halls, ring road, colonnade or cloister), and the corresponding structural method, plane shapes and ratios. Also examines their evolution process from the perspective of time and space in the regional and historical contexts.

This study tries to provide not only a reference frame for the static conservation and dynamic development of the Kumbum Monastery but also a guidance for the future preservation of Tibetan Buddhist buildings in Qinghai.

## Materials and methods

### Study objects and methodology

Based on the results of existing research on Tibetan Buddhist architectural types, the grand halls of Buddhas and Buddhist colleges are the core building types of Tibetan Buddhist monasteries [6, 8, 18]. According to this finding, we considered the spatial functions, religious importance, and the construction frequency in the history of monastery construction as a classification basis to categories building types in Kumbum Monastery and limited the objects to be studied to 12 palace buildings, among 52 buildings of Kumbum Monastery. The 12 palace buildings included the grand halls of Buddhas (8 samples) and Buddhist colleges (4 samples) (Fig 2).

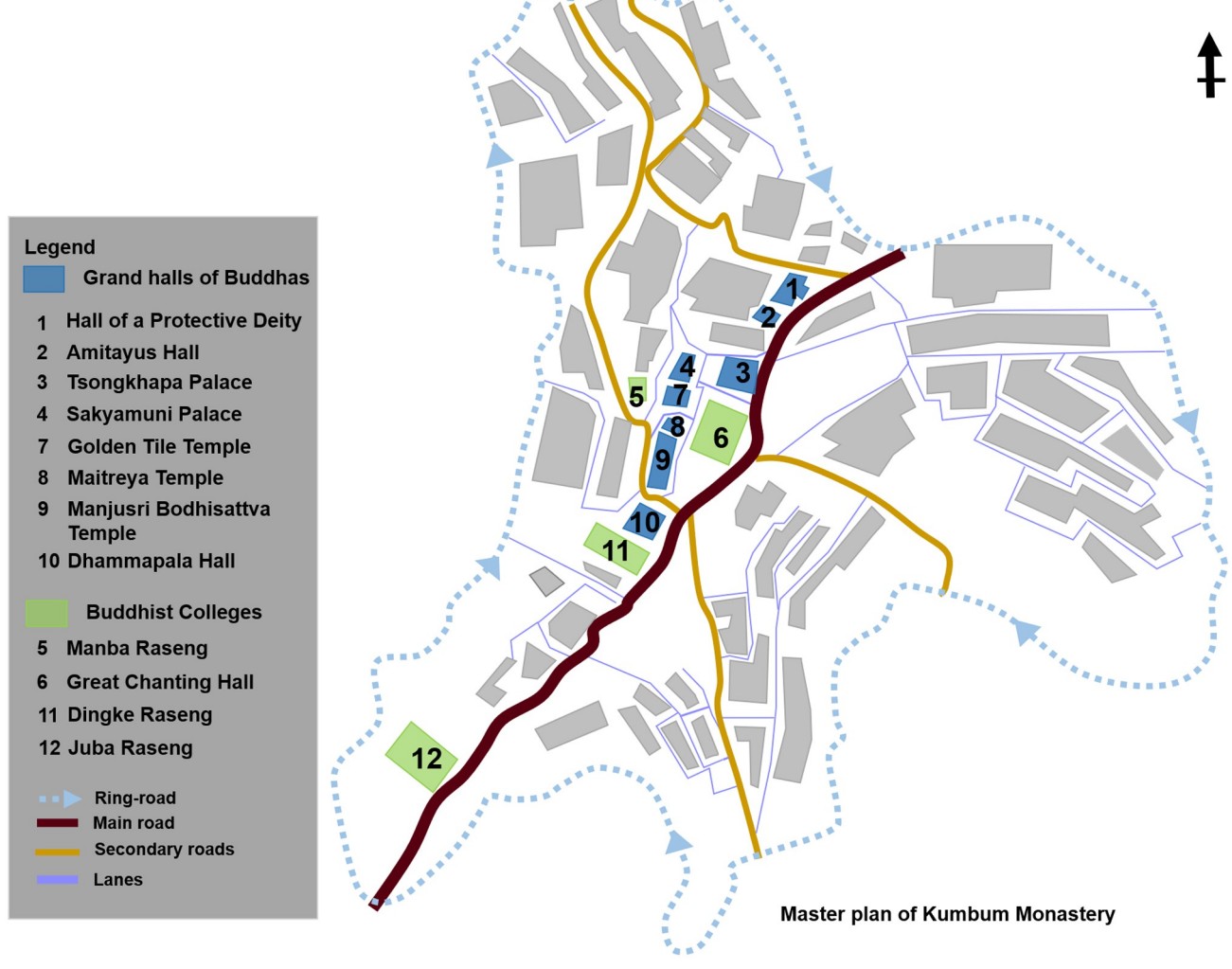

**Legend**

■ **Grand halls of Buddhas**

1 **Hall of a Protective Deity**
2 **Amitayus Hall**
3 **Tsongkhapa Palace**
4 **Sakyamuni Palace**
7 **Golden Tile Temple**
8 **Maitreya Temple**
9 **Manjusri Bodhisattva Temple**
10 **Dhammapala Hall**

■ **Buddhist Colleges**

5 **Manba Raseng**
6 **Great Chanting Hall**
11 **Dingke Raseng**
12 **Juba Raseng**

••▷ **Ring-road**
— **Main road**
— **Secondary roads**
— **Lanes**

**Master plan of Kumbum Monastery**

**Fig 2. Distribution of case studies.**

**Table 1. List of case studies.**

| Architecture Category | Architecture | Foundation Period | Formation Period |
|---|---|---|---|
| Grand Halls of Buddhas | Maitreya Temple | 1577 | 1577 |
| | Tsongkhapa Palace | 1594 | 1594 |
| | Sakyamuni Palace | 1604 | 1604 |
| | Golden Tile Temple | 1622 | 1711 |
| | Dhammapala Hall | 1592 | 1592 |
| | Amitayus Hall | 1717 | 1781 |
| | Hall of a Protective Deity | 1692 | 1826 |
| | Manjusri Bodhisattva Temple | 1592 | 1734 |
| Buddhist Colleges | Great Chanting Hall | 1612 | 1766 |
| | Juba Raseng | 1646 | 1767 |
| | Manba Raseng | 1711 | 1818 |
| | Dingke Raseng | 1817 | 1886 |

Monastery maps, photographs, data from the surveys of the 1992 renovation project, and related history and information were collected from the literature written in the Tibetan and other languages. Field research and online map services were used and classified. Interviews conducted with monks and craftsmen were analysed and incorporated in the analysis map.

The paper then uses time and space as research parameters. In terms of temporal parameter, the palace buildings have been repeatedly repaired, rebuilt, and expanded during the 563 years of their construction. Most of the current palace buildings are different in form from the original buildings. Furthermore, because verifying the images and data of the monastic buildings that were first built is difficult, we used time as a standard when the current structure and form of the main body of the buildings was constructed (Table 1). In terms of spatial parameter, based on the spatial function of the building, the research objects can be divided into grand halls of Buddhas and Buddhist colleges. Then we explore the spatial features through dividing space types due to the spatial elements and combination of spatial elements. Based on this, analyze the corresponding structural processing methods, plane shapes and ratios of each type. In addition, Kumbum Monastery is one of the six main monasteries of the Gelugpa Sect of Tibetan Buddhism, and it was built after the four main monasteries in Tibet. Furthermore, the Gelugpa Sect has a strict and perfect monastic management system and regulations, covering laws, organisations, and monastic buildings [22–24]. Therefore, we compared the spatial features of Tibetan Buddhist palace buildings in Tibet with those of the 12 palace buildings of Kumbum Monastery and analysed them (Fig 3).

With regard to the temporal and spatial features of the evolution process of palace buildings, four factors are considered important for the objectives: 1) the developmental background of the monastery's palace buildings (the development of Tibetan Buddhism in Qinghai, the history of the construction of the monastery's palace buildings, and the evolution of the palace buildings of Tibetan Buddhism in Tibet); 2) the spatial composition type and features of the monastery's palace buildings; and 3) the spatial and temporal evolution of the palace buildings' space of Kumbum Monastery and their development process.

## Results and discussions

### Background overview

According to the design principle of Tibetan Buddhist buildings, the historical context is critical to discuss their temporal and spatial features [8, 19]. As a typical example of Tibetan

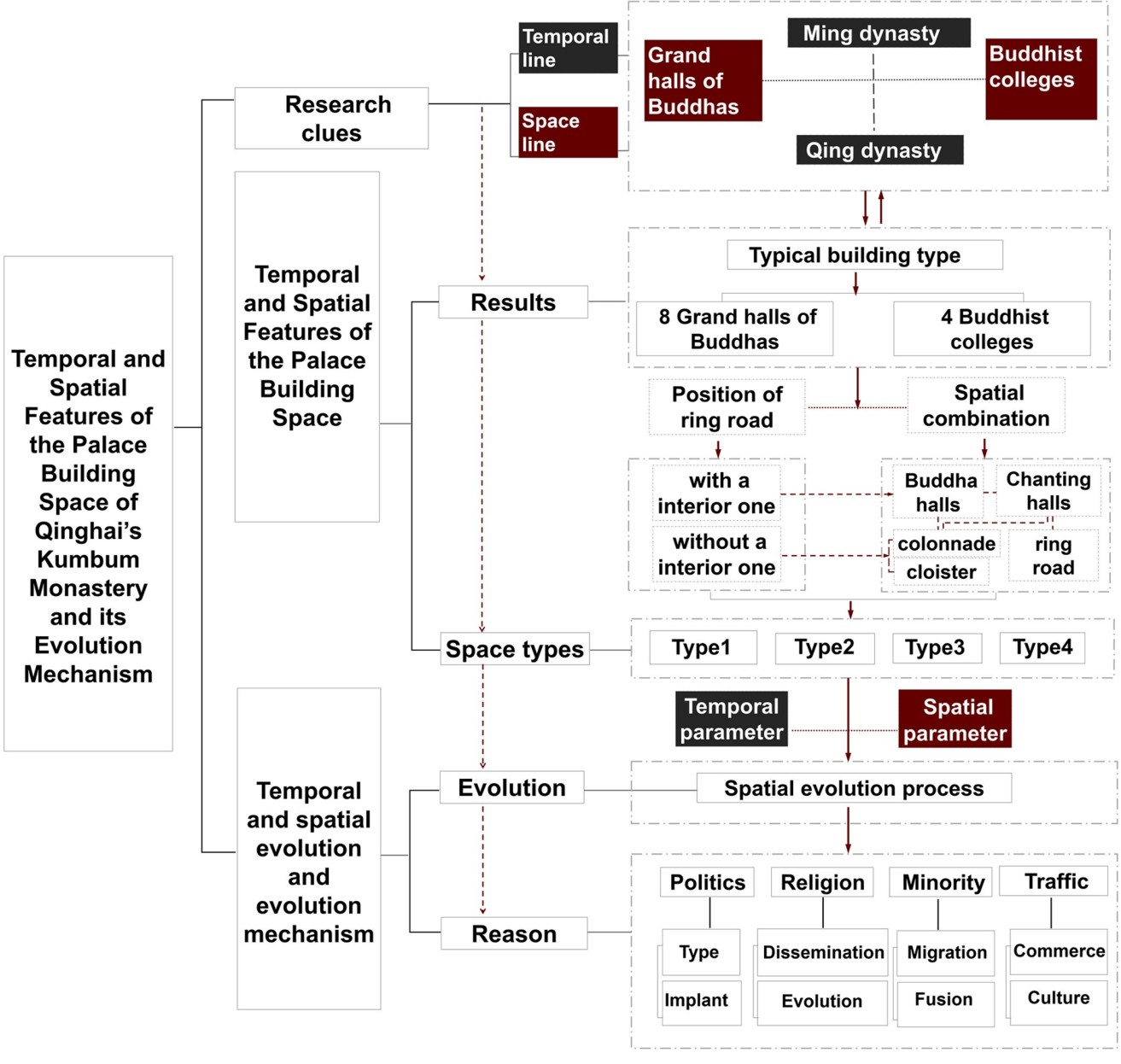

**Fig 3. Research method framework.**

Buddhist monasteries, Kumbum Monastery is located in Qinghai Province, which is away from Tibet, but is the base for the reintroduction of Tibetan Buddhism, the place of the Gelugpa Sect establishment, and the bridgehead and transfer station for the introduction of the Gelugpa Sect into Inner Mongolia. Therefore, the history of the construction of the palace buildings of the monastery and the evolution of the palace buildings of Tibetan Buddhist monasteries in Tibet are the basis and foundation for the study of the spatial evolution of the palace buildings of the monastery, and analysing these buildings can establish a solid foundation for further investigation and classification of the history of the construction of the palace buildings of Kumbum Monastery.

| Social background | Construction category | Period | Case |
|---|---|---|---|
| 1368, the Recruitment Policy of the Ming Dynasty; The formation of Qinghai Hui and Mongolian settlements; the leaders and nobles of various ethnic groups chose to belong to the Ming court; Han immigration due to Military policy;<br>1546, Gelug sect monastery found;<br>1603, organizing system reorganization;<br>1612, Buddhist college lecture system established. | Grand hall of Buddhas(6), Buddhist college(1), Pagoda(1) | I (1379-1629) | 8 |
| 1636, The Qing Dynasty's policy of only allowing the Gelug sect's development;<br>1652, The fifth Dalai Lama is canonized as the leader of Tibetan Buddhism;<br>1723, canonization of the Gelug Sect of Tibetan Buddhism in Qinghai;<br>1724, quelled the rebellion in Qinghai;<br>1727, removed restrictions in Tibetan Buddhism in Qinghai and establish a system of ministers stationed in Tibet. | Grand hall of Buddhas(4), Buddhist college(3), Pagoda(2),auxiliary buildings(7) | II (1649-Mid 18th century) | 16 |
| 1876, the Hui in Qinghai uprising implicated severe destruction of monastery buildings;<br>1912, the Republic of China government was established;<br>1912-1936, a series of laws and regulations about Tibetan Buddhism were promulgated | Buddhist college(4), Grand hall of Buddhas(1), Pagoda(1) | III (1874-1942) | 6 |

**Fig 4. Historical and cultural background of the construction history of Kumbum Monastery.**

**Historical background of the construction of the palace buildings of Kumbum Monastery.** Considering the sociopolitical background, major historical events, and the development of the organisation system of Kumbum Monastery, the construction history of the monastery can be divided into three periods of: building the monastery prototype (1379–1648), completing the Buddhist colleges (1649–1873), and continuing the construction of the monastery (1874–1942) (Fig 4).

The period of building the monastery prototype (1379–1648): Tsongkhapa founded the Gelugpa Sect of Tibetan Buddhism in the 15th century and first constructed the monastery to commemorate its birthplace. Subsequently, several monks visited this place and improved the monastic system. The central government of the Ming Dynasty developed many policies (such as increased the ordaining of Tibetan Buddhist monks, protected of Tibetan Buddhist monastic property, and granted monks specific immunity from crimes) to support the development of the monasteries of the Gelugpa Sect, including Kumbum Monastery. Furthermore, in Qinghai, the Ming dynasty implemented he units of Mongolia, Tusi, and thousand and one hundred household systems for the Mongols, Tibetans, and Han Chinese, respectively, to weaken the ties among various ethnic groups and developed policies to offer amnesty and enlistment to rebels in the minority ghettos. Because the Tibetan Buddhist monasteries in the country owned large lands and monastic tribes, the tribal chiefs pledged allegiance to the Ming government to politically entrench themselves. In the meantime, they cooperated with the monasteries and funded the construction of Kumbum Monastery. Kumbum Monastery was initially built with the concerted support of the Ming dynasty, Tibetan Buddhist groups [25], and the minority chiefs in Qinghai. They built seven halls around the Tsongkhapa Memorial Pagoda. The palace buildings were constructed mainly due to their religious significance and according to the instructions of the great monks and masters.

The period of completing the construction of Buddhist colleges (1649–1873): The Qing dynasty desired to consolidate its rule over Mongolia and Tibet through the leadership of the

Gelugpa Sect in Tibetan Buddhism. The Qing dynasty only praised the Gelugpa Sect highly [26], enabling the sect to gradually gain power to administer Qinghai. Therefore, Kumbum Monastery grew in power, the number of monks surged, and the system and buildings of Buddhist colleges were completed. However, in 1722, the Lobuzangdanjin (the leader of the Mongol tribe in Qinghai) commenced a rebellion against the Qing dynasty, and Kumbum Monastery was implicated in this event due to its long-standing involvement with Mongolian tribal leaders [27]. After the rebellion ended, the Qing dynasty limited the political priorities given to the monastery and only allowed it to focus on religious activities and architectural construction. Consequently, the construction of the monastery buildings was not substantially affected. The main construction projects included seven palace buildings. The Buddhist colleges and their building systems were completed. Since then, the monastery has become one of the main monasteries of the Gelugpa Sect.

The period of continuing the construction of the monastery (1874–1942): The natural economy of rural China, on which the late Qing dynasty depended for its rule, began to collapse; therefore, its power gradually declined. Moreover, because the central government had to protect the country against the invasion of Western countries and suppress peasant rebellions, it had no time to pay attention to the development of Tibetan Buddhist and the construction of the monastery. During the period when the Republican government was replacing the Qing dynasty as the rulers of China, the government intended to adjust the long-term imbalanced spatial pattern of Tibetan Buddhist development in Qinghai, and therefore, constructed additional monasteries in areas outside Hehuang and implemented several initiatives to ensure the economic development of Tibetan Buddhist monasteries (1912) and promote the balanced development of Tibetan Buddhism in Qinghai. Based on this, the focus of development of Kumbum Monastery shifted to the monastic economy. The construction of the monastery mainly focused on the reconstruction and restoration of the original palace buildings and small expansion. The monastery was reconstructed on the original site, and the original forms and structures were used in reconstruction.

In the aforementioned three periods, the development of the monastery has different emphases, but it always gives priority to the grand halls of Buddhas and the Buddhist colleges.

**Evolution of the palace buildings of Tibetan monasteries.** According to the changes in the spatial layout and typical building components (column cap and supporting timber) of buildings, the evolution of the palace buildings of Tibetan monasteries between the 7th and 20th century (i.e. from Tibetan primitive religious period to the period of the Republic of China) was classified into five periods [7] (Fig 5).

The first period (from the 7th to 10th century): a square Buddha hall was placed in the middle, and a left-turned worship path, named as ring road, was built around it. Chanting halls were placed in the rooms that were in front of or on the left and right sides of the Buddha hall and ring road.

The second period (from the late 10th century to the first half of 13th century): the spatial layout of buildings imitated the first period, and the areas of the Buddha hall, ring road, and chanting halls were significantly expanded.

The third period (from the second half of the 13th century to the end of the 14th century): the basic spatial form of the palace building was formed. The palace buildings included the gate court, chanting hall, Buddha hall, and ring road located outside the Buddha hall.

The fourth period (from the 15th century to the 1740s): due to the continuous growth of the number of monks and laymen, who attended church, the indoor ring road debilitated, and the worship path was relocated from around the Buddha Hall to outside the palaces.

The fifth period (from the 1750s to the 20th century): since the mid-17th century, when the Gelugpa Sect was dominant among all the sects, the dominant space in monastery buildings,

| Stage of development | Illustrating picture | Representative architectural examples |
|---|---|---|
| The first period<br><br>(7th - 10th centuries) | | Grand Hall of Buddhas of Ra-mo-che Temple |
| The second period<br><br>(late10th century- first half of 13th century) | | Great Hall of Zhatang Monastery |
| The third period<br><br>(the second half of the 13th century -the end of the 14th century) | | Grand Hall of Buddhas of Changzhu Monastery |

**Fig 5. Evolution of palace buildings in Tibetan monasteries (from the 7th to 20th century).**

that is, the form of 'Dugang Ceremony' (the Buddha hall where monks gather and chant sutra; 'Dugang Ceremony' is the practice of constructing palace buildings, which is a programmed system that expresses the religious mood based on the Tantric mandala of Tibetan Buddhism.), was mature (Zhang 2016). Its form is characterised by the following aspects: 1) the palace expands in area. The monastic hierarchy is reflected by the number of pillars in the chanting hall, and 2) the area of the chanting hall increased to solve the spatial need resulting from the increase in the number of monks, thus the area of the Buddha halls decreased.

## Spatial composition types and features of the palace buildings

By analysing the spatial layout of the building, we can identify the spatial composition rules and forms associated with environmental and topographical conditions and human relations in society. According to the aforementioned evolution of palace buildings in Tibetan monasteries, the spatial evolution of monasteries in Tibet is reflected in the location of the ring road and the spatial combination of Buddha halls and chanting halls. The 12 palace buildings of Kumbum Monastery will be divided into four types in our next study on the basis of the position of the ring road and spatial combination characteristics of the dominant spaces in the buildings. Moreover, we will discuss the spatial and temporal features of the Kumbum Monastery buildings by comparing the evolution of the monastery buildings with that of the buildings in Tibet.

**Type 1: The ring road surrounding Dugang-style palace space.** This type refers to the temple space with an interior ring road around the independent grand halls of Buddhas used

for placing Buddha statues and other sacred objects at the centre. They were constructed in the period between the Ming Dynasty and the early Qing Dynasty (1577–1604). The building types are the grand halls of Buddhas. In the form of colonnades or eaves galleries shaped as the letter 'U' or the Chinese character '回' (round and cycle). The width-to-depth ratio of the hall is equal or approximately equal to 1, that is, the plan tends to be square (Fig 6). At the same time, the approach of 'moving pillars' or 'removing pillars' is used in the grand halls of Buddhas to place the statue of Buddha or the pagoda at the centre. The 'moving pillar' approach, that is, moving the inner grooves of four pillars outwards by 40 cm, is used in the Maitreya Temple; the approach of 'removing pillars' is used in the Tsongkhapa Palace, the Sakyamuni Palace, and the Dhammapala Hall, thus making the rooms in these palaces free of pillars. In the column grid, the size of Mingjian (the space between the two pillars in the middle of the plane in the traditional architecture of ancient China) generally does not exceed 5 m, Cijian (the pillars next to Mingjian decreasing progressively) is short, and Shaojian (the pillars next to Cijian with decreasing progressively size) is the shortest (Table 2). This type of palace buildings comprises a single-story or two-story cloister, Han-style gable, and hip roof or East Asian hip-and-gable roof. The buildings of this category were mainly built in the first stage of the construction of Kumbum Monastery (1379–1643). This type of palace space is in line with the fourth period of the evolution of the palace buildings of Tibetan Monastery (the 15th century–1740s). This type of buildings was built around the first construction period, Tsongkhapa Memorial Pagoda, and was distributed in a linear fashion in the central area of the monastery.

**Type 2: Single-sided eaves of the Duguang-style palace space.** This type of the palace space indicates that only the main spaces, which are connected with a unilateral colonnade and without the interior ring road, are used for placing Buddha statues and other scared objects. They were constructed between 1711 and 1734. The types of building are the grand halls of Buddhas. Based on the construction pattern of the palace buildings in the first period,

| Architecture | Plan schematic diagram | Section schematic diagram | W：D | Foundation Period | Formation Period |
|---|---|---|---|---|---|
| Maitreya Temple | | | 1：1 | Ming Dynasty (1577) | Ming Dynasty (1577) |
| Dhammapala Hall | | | 3：4 | Ming Dynasty (1592) | Ming Dynasty (1592) |
| Tsongkhapa Palace | | | 4：5 | Ming Dynasty (1594) | Ming Dynasty (1594) |
| Sakyamuni Palace | | | 3：4 | Ming Dynasty (1604) | Ming Dynasty (1604) |

**Fig 6. Spatial features of the palace buildings of type I.** (W, width of the Palace Halls; D, depth of the Palace Halls).

**Table 2. Dimensions of the planar column network of the palace space for type I.**

| Architecture | W1[a](m) | W2[b](m) | W3[c](m) |
|---|---|---|---|
| Maitreya Temple | 4.000 | 2.900 | 1.600 |
| Tsongkhapa Palace | 3.600 | 3.200 | 1.600 |
| Sakyamuni Palace | 3.500 | 3.200 | 1.350 |
| Dhammapala Hall | 4.160 | 4.150 | 3.850 |

[a]W1, the Width of Mingjian.

[b]W2, the Width of Cijian.

[c] W3, the Width of Shaojian.

their space gradually expands to the south and continues the original linear spatial structure. The buildings are shaped similar to the Chinese character '凸' and comprise the front corridor, unilateral colonnade, and Buddha Hall, with single-side colonnades connected to the independent Buddha Hall. The width-to-depth ratio of the Buddha Hall is 7/5–3, that is, the Buddha Hall is rectangular (Fig 7). The centre of the hall is the sanctuary space where the sacred objects are placed, and the space height depends on the height of the sacred objects. For example, the Golden Tile Temple has three levels, with 20 cubic columns located on the ground floor of the hall, and 6 corridor columns and 24 columns embedded in the walls. To accommodate the large volume of the Great Silver Pagoda, pillars on each level of the hall are arranged according to the 'removing pillar' approach, that is, the four pillars in the middle are removed, thereby forming a patio with the connected top and bottom (Fig 8, S1 Fig). This type of the palace building has a gable and hip roof and flush gable roof. These buildings were built mainly at the end of the first period of the construction of Kumbum Monastery (1379–1643). The palace space form changed partially compared with the form of the fourth period buildings (from the 15th century to the 1740s). For example, the construction of the ring roads in Manjusri Bodhisattva Temple was cancelled.

**Type 3: Three-stage palace space with cloisters connecting to Dugang.** This type of the palace space indicates that only the main spaces, which are connected with a cloister and

| Architecture | Plan schematic diagram | Section schematic diagram | W：D | Foundation Period | Formation Period |
|---|---|---|---|---|---|
| Golden Tile Temple | | | 7：5 | Ming Dynasty (1622) | Qing Dynasty (1711) |
| Manjusri Bodhisattva Temple | | | 3：1 | Ming Dynasty (1592) | Qing Dynasty (1734) |

**Fig 7. Spatial features of the palace buildings in type II.** (W, width of the Palace Halls; D, depth of the Palace Halls).

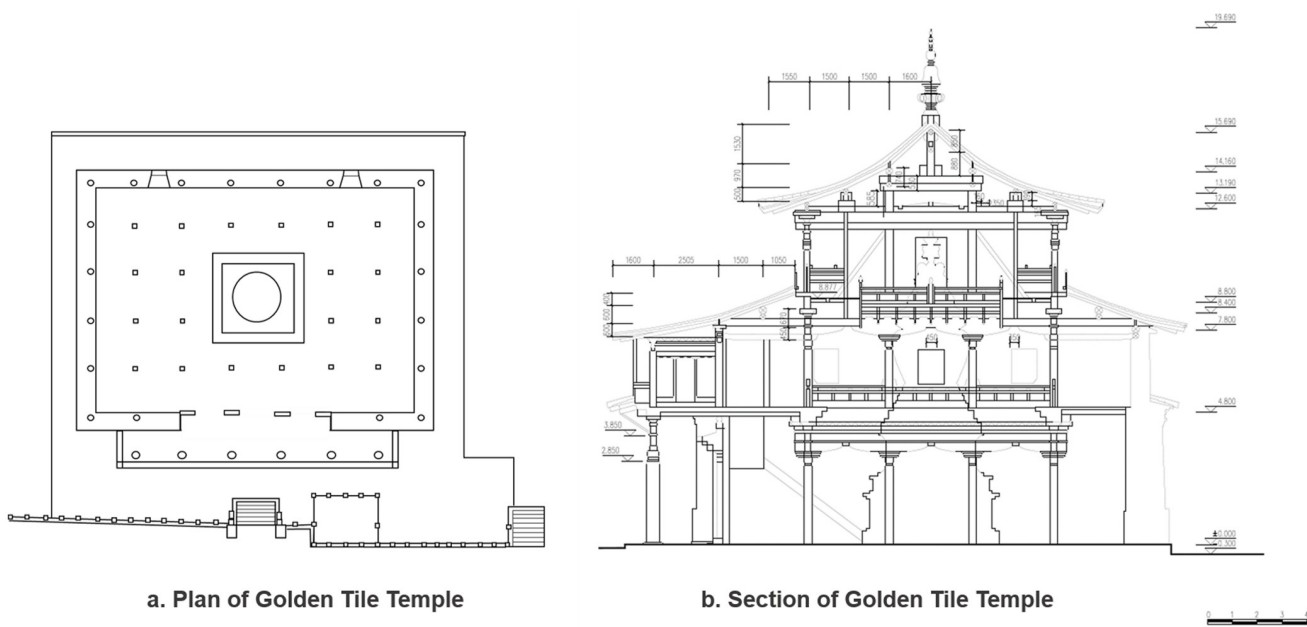

a. Plan of Golden Tile Temple

b. Section of Golden Tile Temple

**Fig 8. Plan and section of the Golden Tile Temple.**

without the interior ring road, are used for placing Buddha statues and other scared objects. They were constructed between 1766 and 1886. All the buildings of this type are Buddhist colleges (4 samples). These buildings are rectangular in shape, and their width-to-depth ratio of the chanting hall ranges from 5/6 (0.83) to 13/10 (1.3). The buildings are shaped as the Chinese character '凸' (2 samples) or are rectangular in shape (2 samples) (Fig 9). The buildings comprise an entrance, a high corridor, a courtyard, a cloister, and a chanting hall/Buddha Hall. The main building of these halls is a two-story, flat-topped, and Tibetan-style building. The ground floor serves as a chanting hall and porch; the second floor forms a patio shaped as the Chinese character '回' or half of '回' on top of the chanting hall, with small Buddha halls, sutra storage, and auxiliary rooms such as the Great Chanting Hall (Fig 10, S1 Fig) around it. The U-shaped ring corridors or monk rooms are situated in front of the chanting hall of the three buildings of the Buddhist colleges, namely the Great Chanting Hall, Juba Raseng, and Manba Raseng. Three sides of the front courtyard are double-layered flat-roofed cloisters. A double-opening gate is present in the middle of the east wall, and a gable and hip roof pavilion are available on top of the flat roof of the gate. The perfect example of this building type is the Great Chanting Hall (Fig 11, S1 Fig). The buildings of this type were developed mainly in the third stage of monastery construction (1649–1873). Kumbum Monastery could only be expanded towards the north and south due to the terrain constraints; the monastery is located in the Lotus Mountain, which comprises eight north–south mountain bays. Its spatial features correspond to the fifth period (from the 1750s to 20th century) of the evolution of the palace buildings of Tibetan Monastery, that is, expanding the area of the chanting hall and shrinking the area of the Buddha hall. The number of pillars used in the chanting halls reflects the hierarchy of monastery construction. The system of the Buddhist colleges of Kumbum Monastery and the time when they started to be developed are based on the educational system and the order of construction of the four Buddhist college buildings established by the Gelugpa Sect of Tibetan Buddhism.

| Architecture | Plan schematic diagram | Section schematic diagram | W：D | N | Foundation Period | Formation Period |
|---|---|---|---|---|---|---|
| Great Chanting Hall | | | 13:10 | 108 | Ming Dynasty (1612) | Qing Dynasty (1766) |
| Juba Raseng | | | 1：1 | 36 | Qing Dynasty (1646) | Qing Dynasty (1767) |
| Manba Raseng | | | 7：6 | 30 | Qing Dynasty (1711) | Qing Dynasty (1818) |
| Dingke Raseng | | | 5：6 | 20 | Qing Dynasty (1817) | Qing Dynasty (1886) |

**Fig 9. Spatial features of the palace buildings of type 3.** (W, width of the Palace Halls; D, depth of the Palace Halls; N, the number of pillars).

**Type 4: Other variants.** In addition to the aforementioned three types, there are two other variants of the palace building in Kumbum Monastery, both of them were built without an interior ring road, and the Buddha halls were linked with courtyard initially or a U-shaped closed corridor (Fig 12).

The first variant is the Hall of the Protective Deity. It was originally built in the Dongla Mountain outside the monastery and moved to the present site in 1692. Its original spatial form did not have the double-layered cloister in front of the temple (it was added in 1826) and the gilded copper tile temple roof (it was constructed in 1809). The Buddha Hall is rectangular, its width-to-depth ratio is 7/3, and the main hall has two floors.

The second variant is the Amitayus Hall. It was built in 1717 to pray for the longevity of the Dalai Lama stationed in Kumbum Monastery at that time. The Hall is rectangular, and the

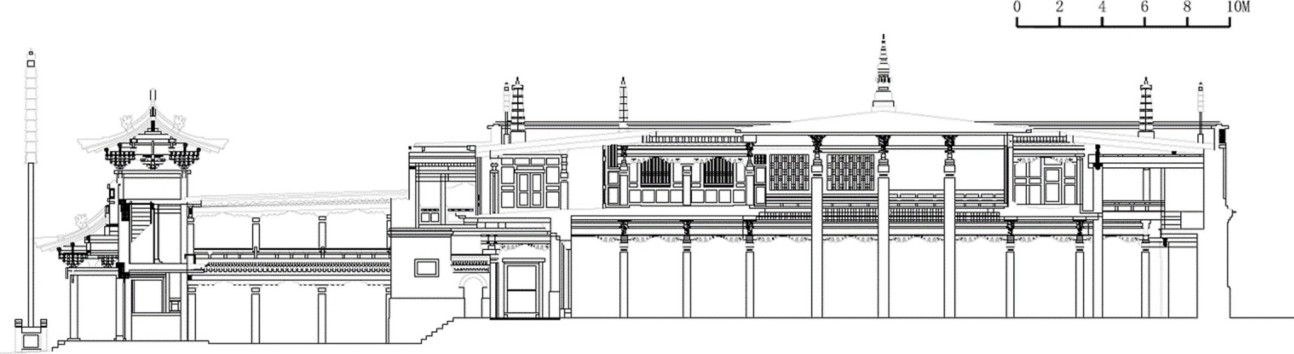

**Fig 10. Section of the Great Chanting Hall.**

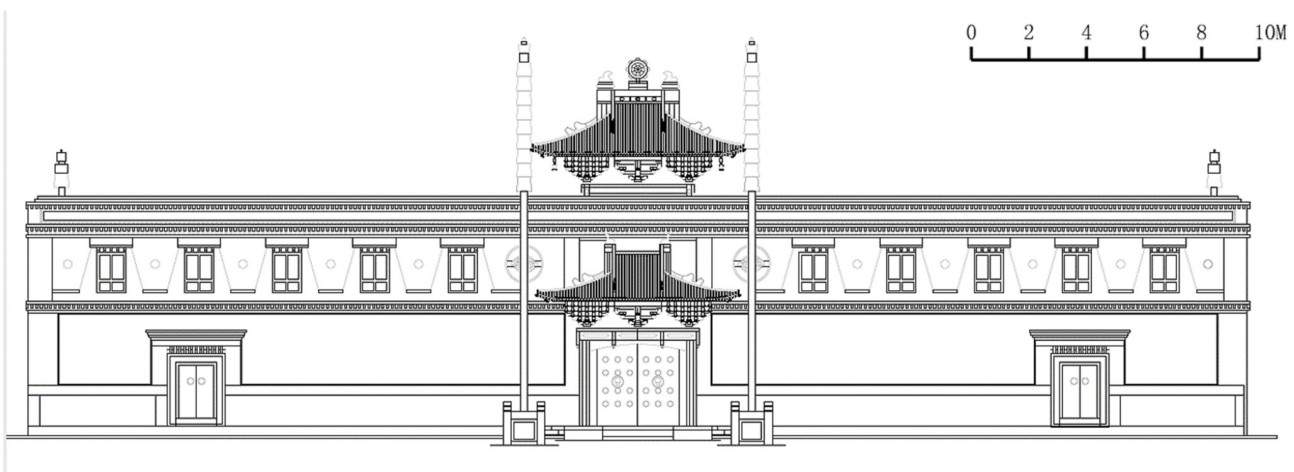

**Fig 11. Elevation of the Great Chanting Hall.**

width-to-depth ratio is 5/3. The centre of the Hall is a well-hole space, and the second floor is surrounded by small cloisters. Colonnades are present in the Mingjian and Cijian of the front elevation, and a U-shaped closed corridor is situated in the back, left, and right. The Amitayus Hall has a two-story, pavilion-type East Asian hip-and-gable roof.

The main halls of both the variants were completed in the beginning of the third stage of the construction of Kumbum Monastery (1649–1873). The original form of the grand halls of Buddhas is in line with the features of the fourth stage of the evolution of the palace buildings in Tibetan monasteries, that is, discontinuing of the indoor ring road. Moreover, the practice of adding the cloisters is in line with the spatial combination elements and patterns of the other palace buildings (the four Buddhist colleges) built in the third period of Kumbum Monastery construction.

| Architecture | Plan schematic diagram | Section schematic diagram | W: D | Foundation Period | Formation Period |
|---|---|---|---|---|---|
| Amitayus Hall | | | 5: 3 | Qing Dynasty (1717) | Qing Dynasty (1781) |
| Hall of a Protective Deity | | | 7: 4 | Qing Dynasty (1692) | Qing Dynasty (1826) |

**Fig 12. Spatial features of the palace buildings of type 4.** (W, width of the Palace Halls; D, depth of the Palace Halls).

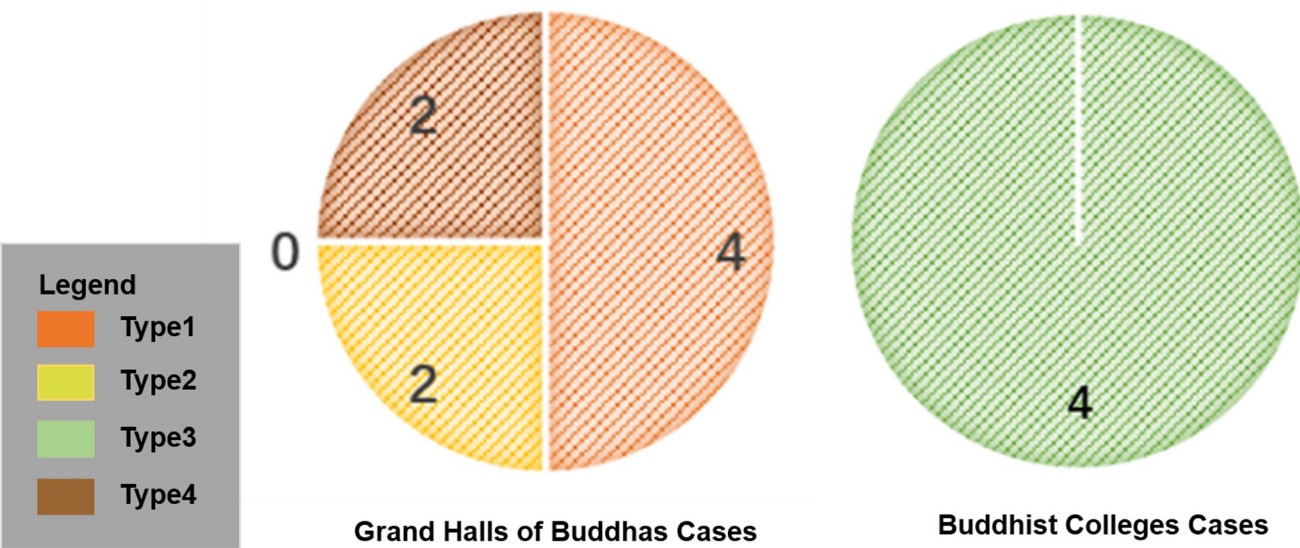

**Fig 13. Examples of the four types of palace spaces in the buildings of the grand halls of Buddhas and Buddhist colleges.**

### Temporal and spatial evolution

Based on the aforementioned spatial types of the palace buildings, we explored the spatial features of the palace buildings of Kumbum Monastery and its evolution process in the temporal and spatial dimensions.

**Spatial and temporal evolution and features of the palace buildings.**    The spatial and temporal evolution process and features of the palace buildings in Kumbum Monastery can be discussed from three aspects: architectural type, building distribution, and spatial composition.

First, among the spatial forms of the palace buildings of Kumbum Monastery, type 1 (the ring road surrounding the Dugang-style palace space), type 2 (single-sided eaves of the Duguang-style palace space), and all other variants appear in the Buddhist halls. Type 3 (three-stage palace space with cloisters connecting to Dugang) is found only in the Buddhist colleges (Fig 13).

Second, type 1 and 2 are concentrated in the central area of the monastery, and type 3 is distributed in both the central and peripheral areas. Type 4 is only scattered in marginal area of the monastery nearby the entrance (Fig 14).

Third, there are four palace buildings of type 1, as well as type3, which is twice the number of buildings of type 2 or type 4 (Fig 15). It corresponds to the rise and fall of Kumbum Monastery construction activities and the historical context. And the width-to-depth ratio of the main hall includes 1(type1), 7/5-3 (type 2), 5/6-13/10 (type 3), and 7/3-5/3, it shows that the plane of the main hall follows the rectangular form, but the specific proportions are not regulr.

Fourth, the spatial composition of the palace buildings reflects the characteristics of changes with the construction space; the characteristics are manifested in the following two points: 1) Among the palace buildings of Kumbum Monastery, the building with the courtyard connected to the entrance of Dugang space was built in the late Qing Dynasty, along with its four Buddhist college and two Buddha hall buildings. The main body of the Buddhist college and Buddha hall buildings was formed after 1766 and 1767, respectively. The buildings formed before them were freestanding Buddha hall buildings. 2) Among the four Buddhist college

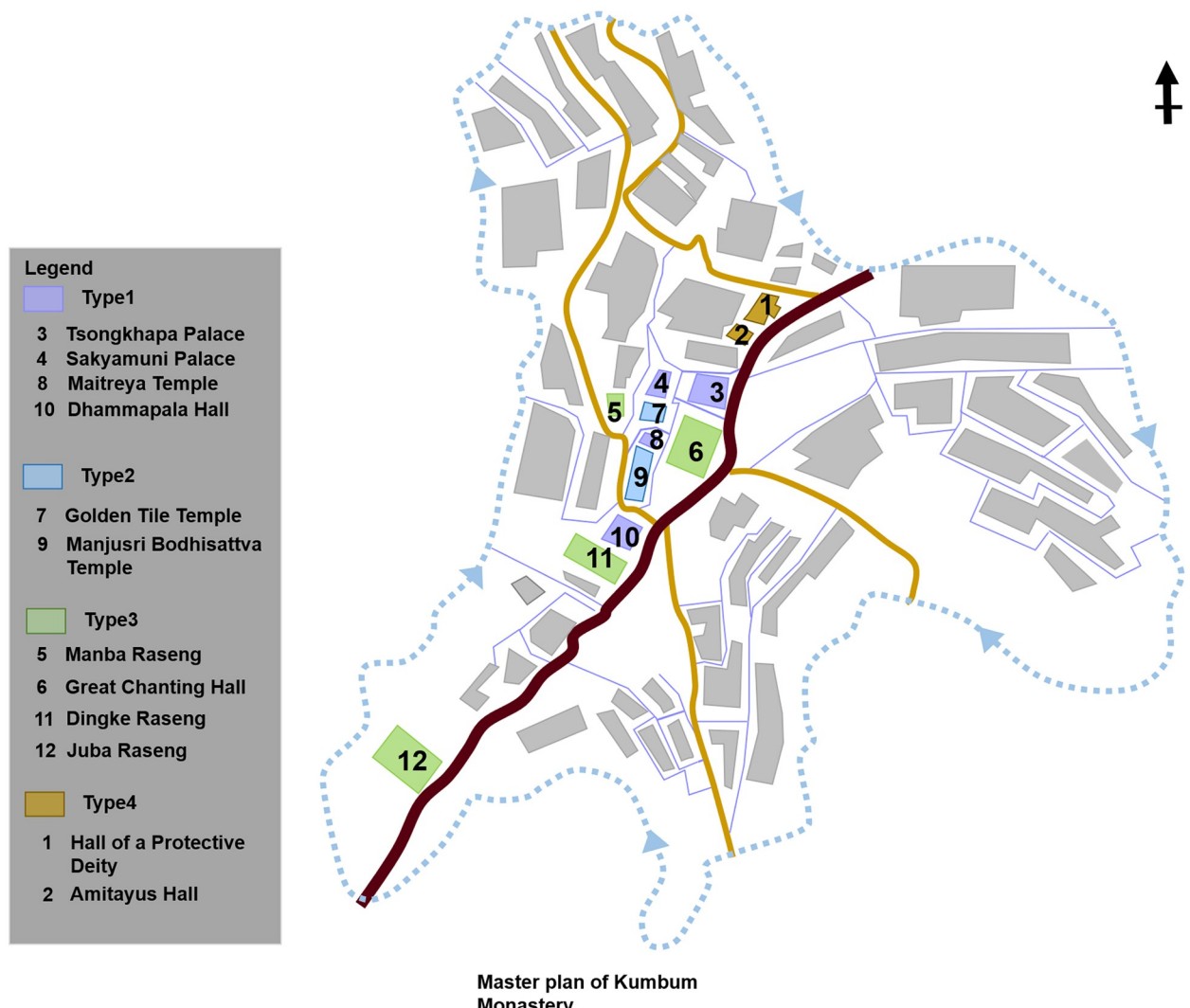

**Legend**

- **Type1**
  - 3 Tsongkhapa Palace
  - 4 Sakyamuni Palace
  - 8 Maitreya Temple
  - 10 Dhammapala Hall

- **Type2**
  - 7 Golden Tile Temple
  - 9 Manjusri Bodhisattva Temple

- **Type3**
  - 5 Manba Raseng
  - 6 Great Chanting Hall
  - 11 Dingke Raseng
  - 12 Juba Raseng

- **Type4**
  - 1 Hall of a Protective Deity
  - 2 Amitayus Hall

**Master plan of Kumbum Monastery**

**Fig 14. Distribution of 4 types of palace buildings in Kumbum Monastery.**

buildings, the Great Chanting Hall, Juba Raseng, and Manba Raseng have similar spatial composition forms, that is, there is certain symmetry in their overall layout. Their main building was constructed between the second half of the 18th century and the early 19th century (1766–1818). The Dingke Raseng, which was built in the second half of the 19th century (1886), adds an asymmetrical spatial courtyard layout on the basis of the spatial form of the first three. The earth walls without windows surround the Dingke Raseng, the door is in the middle of the south wall, and the door is arranged along the axis or away from the centre. This courtyard layout, namely Zhuangke-style courtyard (the type of dwelling popular in the eastern part of Qinghai Province and 11 counties and cities along the Yellow River and Huangshui River), serves as the typical residential system in the eastern part of Qinghai.

**Evolution of the spatial structure of the palace buildings.** To analyse the reasons for the evolution of the aforementioned forms of space construction, we should review the historical background of the construction of the palace buildings of Kumbum Monastery, investigate the evolution of the spatial features of the monastery's buildings from the perspective of politics,

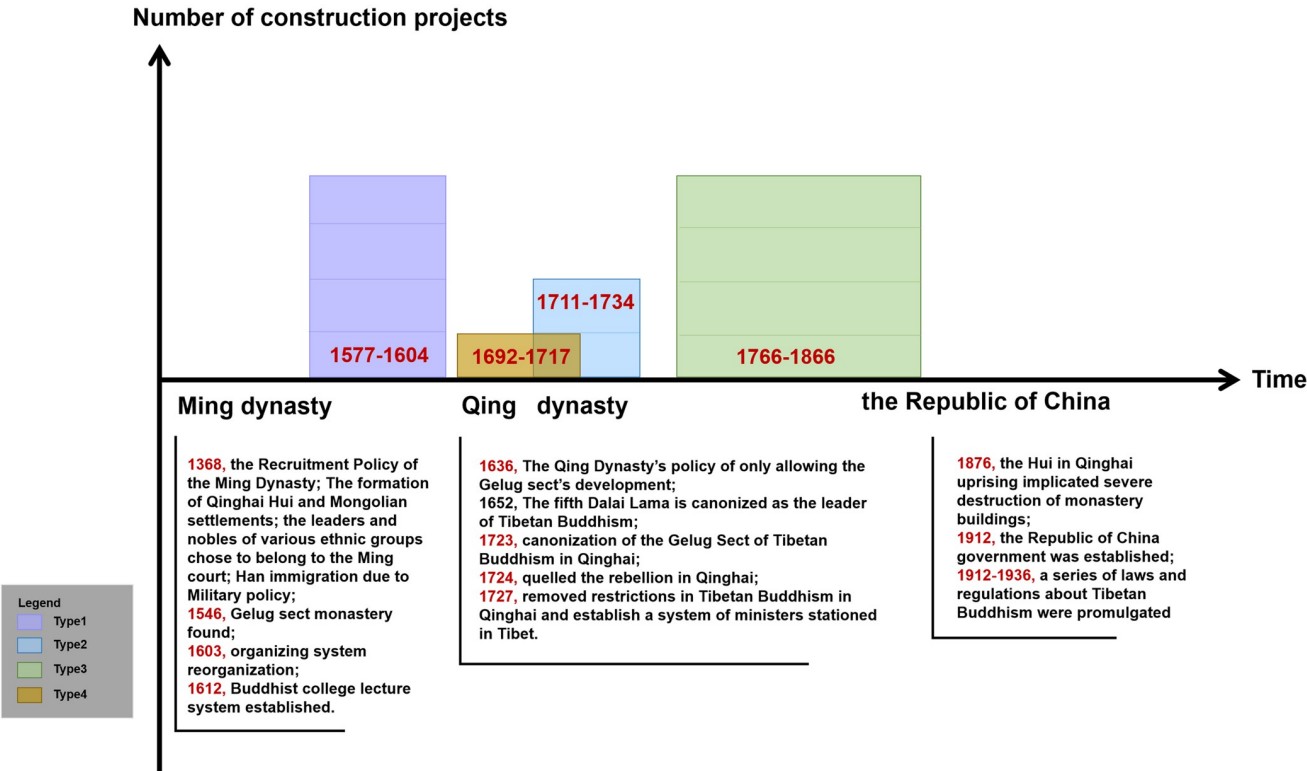

**Fig 15. Evolution of 4 types of palace buildings in Kumbum Monastery.**

religion, nation, and transportation, and explore the dynamic factors causing the spatial features of the palace buildings and the manner in which the multi-culture works.

Types 1 and 2 were constructed in the first stage of the monastery's construction. Since 841, Buddhist monks fled to the Heghuang region of Qinghai to revive Tibetan Buddhism because Tibet adopted the policy of 'eliminating Buddhism' [28]. By the end of the Yuan dynasty (1271–1368), Tsongkhapa came to Qinghai and established the Gelugpa Sect. During the Ming Dynasty (1522–1566), the 11th emperor of the Ming Dynasty encouraged all the sects of Tibetan Buddhism that had constructed monasteries in the Huanzhong region of Qinghai [29]. With the concerted support of the central government of the Ming dynasty, religious groups, and the ruling class of ethnic groups, the palace buildings of Kumbum Monastery were constructed under the instruction of the senior monks to establish their own religious significance. Therefore, the spatial features of the buildings follow the practice of the fourth stage of the evolution of the palace buildings in Tibet, that is, the worship path was moved from the periphery to the outside of the Buddha Hall. With the passage of time, the road changed partly in form, that is, the ring road was abolished. In addition, during the Ming dynasty, the central government organised large-scale migrations towards the northern regions to develop the areas devastated by the war, including the Hehuang of Qinghai, where the Kumbum Monastery was located [30]. Consequently, the construction technology of Central Plains (a traditional regional concept of the Han nationality, which refers to a vast area with Henan Province as the core that extends to the middle and lower reaches of the Yellow River, and regarded as the birthplaces and centre of Chinese ancient civilization.) of China was introduced to the area [31] and thus the style of the buildings in the Yellow River basin

gradually became similar to that in Central Plains. To place the large Buddha statue and Holy Tower in the centre of the palace building in Kumbum Monastery, types 1 and 2 adopted the structural layout of 'removing pillars' or 'moving pillars' in Chinese style in the Central Plain area of the Ming dynasty.

Type 3 is observed in all the Buddhist colleges of Kumbum Monastery. Before the construction of the Buddhist colleges of Kumbum Monastery was completed, the Buddhist colleges and their architectures were already developed in a fixed form in four dominant and largest monasteries of the Gelugpa Sect located in Lasha. Although Kumbum Monastery, as the fifth main monastery of the Gelugpa Sects, was built outside Tibet, its Buddhist colleges system was formed in the Qing dynasty. In 1652, the emperor of the Qing Dynasty established the regency system in Tibet. He appointed the Gelugpa Sect as the leader of various sects of Tibetan Buddhism and acquiesced to the conversion of other sects into the Gelugpa Sect. This led many monasteries of other sects in the Qinghai to convert into the Gelugpa Sect. Moreover, 31 monasteries converted and turned into the subordinate temples of Kumbum Monastery. In this social context, Kumbum continued to follow the practices of the organization of Buddhist college, thereby establishing the order and spatial characteristics of architectures of the four dominant Gelugpa Sect monasteries in Lasha except for Dingke Raseng because although the other three Buddhist colleges had formed by 1711, Dingke Raseng was the last one to be constructed, and its architectures were formed in 1817. Before its construction, in the first year of the reign of Emperor Yong Zhen of the Qing dynasty (1723), Lobsangdanjin incited the religious forces of the Tibetan Buddhism monasteries in Qinghai to launch an open rebellion, and Chahannuomen (the great lama of Kumbum Monastery) was involved in it. Subsequently, the Qing dynasty's rebellion was quelled. Because the Gelugpa Sect forces were involved, the Qing dynasty accepted the suggestions, presented in Thirteen Suggestions for Dealing with Qinghai Matters, to restrict the development of Gelugpa Sect monastery in Qinghai [32]. Dingke Raseng, which was built after this time, took the initiative to absorb the regional elements of architectural space in Qinghai to prove their legitimacy.

According to the aforementioned results, with influences from the religious dissemination and sociopolitical influence of the Ming dynasty, Kumbum Monastery received the coordinated support of the central government, religious groups, and the ruling class of ethnic groups because the Han immigrants were integrated with the original ethnic groups. On the premise of following the practice of the Tibetan area in the same period, the space of the buildings of Kumbum Monastery has undergone partial changes with time. Simultaneously, it started to adopt the column-net layout technique of Central Plains. Subsequently, driven mainly by the social and political climate of the Qing dynasty and influenced by the leadership of the Gelugpa Sect among the Tibetan Buddhism sects, the Buddhist college system in Kumbum Monastery was gradually completed. All the palace buildings were built according to the practices of the four major Gelugpa Sects in Lhasa. To eliminate the negative influence of the rebellion, Kumbum Monastery took the initiative to imitate the architectural layout of the buildings in Qinghai, indicating that the monastery wanted to take its cue from the Han culture to proclaim its legitimacy to the central government.

Based on this study, typical cases of each typology are worth focusing on in future studies regarding Tibetan Buddhist monastery, with the integration of politics, religion, minority, and traffic to explore the overall evolutionary law of this monastery's buildings. And in respect to architectural preservation and conservation, it should be recognized that the protecting and conserving activities for palace buildings in Kumbum Monastery cannot be constructed according to the same template. In terms to protecting measures, the spatial element, configuration, function and the cultural connotation should be analysed dialectically, then the parts with specific and important cultural value could be clarified.

## Conclusions

Due to the influence of the two strong cultures, namely the Tibetan and Han culture, the architectural space of Kumbum Monastery followed the Tibetan practice and gradually incorporated the spatial features of the Central Han area and the local buildings of Qinghai with the changing political climate, thus producing four types of palace building space, whose spatial and temporal evolution characteristics reflect the process of interpretation of the same religious mandala through different forms of palace buildings and show the religion to politics oriented evolution of the buildings of Kumbum Monastery. Nevertheless, it should be noted that Tibetan Buddhist cannot be limited formal terms. To avoid misuse architectural patterns and simple duplication of Tibetan Buddhist cultural imagery, the spatial configuration and evolution process should be explored, and the essence of cultural identity and religious thoughts should be further discussed and truly understood with the historical context in each project.

## Supporting information

**S1 Fig. Figure of surveying and mapping of Golden Tile Palace.**
(DWG)

**S2 Fig. Figure of surveying and mapping of Great Chanting Hall.**
(DWG)

## Author Contributions

**Conceptualization:** Jing Zhang.

**Data curation:** Jing Zhang.

**Formal analysis:** Jing Zhang.

**Funding acquisition:** Yunying Ren.

**Investigation:** Jing Zhang.

**Methodology:** Jing Zhang, Yunying Ren.

**Project administration:** Jing Zhang.

**Resources:** Jing Zhang.

**Software:** Jing Zhang, Xiaofan An.

**Writing – original draft:** Jing Zhang.

**Writing – review & editing:** Jing Zhang.

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
