## [Decision Letter · Decision Letter 0]

23 Sep 2021

PONE-D-21-25675Temporal and Spatial Features of the Palace Building Space of Qinghai's Kumbum Monastery and its Evolution MechanismPLOS ONE

Dear Dr. Zhang,

Thank you for submitting your manuscript to PLOS ONE. After careful consideration, we feel that it has merit but does not fully meet PLOS ONE’s publication criteria as it currently stands. Therefore, we invite you to submit a revised version of the manuscript that addresses the points raised during the review process.

We look forward to receiving your revised manuscript.

Kind regards,

Jun Yang

Academic Editor

PLOS ONE

Journal Requirements:

2. In your manuscript, please provide additional information regarding the specimens used in your study. Ensure that you have reported specimen numbers and complete repository information, including museum name and geographic location.

For more information on PLOS ONE's requirements for paleontology and archaeology research, see https://journals.plos.org/plosone/s/submission-guidelines#loc-paleontology-and-archaeology-research.

3. We note that Figures 1 and 3 in your submission contain [map/satellite] images which may be copyrighted. All PLOS content is published under the Creative Commons Attribution License (CC BY 4.0), which means that the manuscript, images, and Supporting Information files will be freely available online, and any third party is permitted to access, download, copy, distribute, and use these materials in any way, even commercially, with proper attribution. For these reasons, we cannot publish previously copyrighted maps or satellite images created using proprietary data, such as Google software (Google Maps, Street View, and Earth). For more information, see our copyright guidelines: http://journals.plos.org/plosone/s/licenses-and-copyright.

a. You may seek permission from the original copyright holder of Figures 1 and 3 to publish the content specifically under the CC BY 4.0 license.  

4. We note that Picture 1 in your submission contain copyrighted images. All PLOS content is published under the Creative Commons Attribution License (CC BY 4.0), which means that the manuscript, images, and Supporting Information files will be freely available online, and any third party is permitted to access, download, copy, distribute, and use these materials in any way, even commercially, with proper attribution. For more information, see our copyright guidelines: http://journals.plos.org/plosone/s/licenses-and-copyright.

a. You may seek permission from the original copyright holder of Picture 1 to publish the content specifically under the CC BY 4.0 license. 

Additional Editor Comments (if provided):

Reviewer 1

The authors used typological approaches and numerical method to analyse spatial features of palace buildings of Kumbum Monastery, classified them into four types, and discussed the evolution of their features to obtain the development mechanism of the palace buildings of Kumbum Monastery. The research is valuable. However, there are still some problems:

Structure. Literature Review had better be separated from the Introduction and make a separate section. Study Objects and Methodology should be placed in a separate section of “Methodology”. Background Overview is the content about the study area. It should also be in the “Methodology” section. The above is only the suggestion, please adjust the structure according to the actual situation.

Table. Please pay attention to the format. For example, 3-line table is usually required in published articles.

2.1 Development Background of Tibetan Buddhism in Qinghai. This part is not the key content, and it is too long now. It is suggested refining.

The reference format should be revised according to the journal standard.

The clarity of all figures needs to be improved, and some formats are wrong as shown in Fig.2. Please provide the original figures in the attachment.

In "4.1 Spatial and temporal Evolution and features of the Palace Buildings", I hope to see the evolution of the features of palace Buildings on a map/diagram, but you only gave the pie chart. In view of “Spatial and temporal evolution”, considering adding meaningful maps or modifying this subtitle.

Also, the language needs to be polished by at least one native speakers.

Reviewer 2

The manuscript focuses on spatial features and evolution mechanism of buildings in Kumbum monastery. The topic is interesting, the study area is relevant; consequently, the manuscript can attract many readers in this field; however, a thorough revision is unquestionably needed before acceptance to improve the scientific quality. See my comments below:

1. The structure of the article is chaotic and the writing pattern is not scientific enough. The current structure is difficult to read and cannot form a smooth logical system. It is hoped that the manuscript can be reorganized in the order of Introduction - Methods - Results and discussion - Conclusion;

2. The introduction of research methods is relatively thin. The rationality of the method is the guarantee of the scientificity of the article. The author should introduce the use methods of various materials in detail, while the current introduction is too general. This paper has almost no discussion on typology. I don't know how the author uses the typological method to analyze. It is speculated from the article that the author should want to introduce the morphological evolution of buildings in Kumbum Monastery with the method of process typology, but it is lack of accurate definition. In addition, how is the importance of numerical analysis reflected in the research？ I think the author should strengthen the interactive analysis of this data and the evolution mechanism of building types, not just a simple list.

3. I think section 2, 3 and 4 of the article can be integrated for in-depth discussion without parallel listing. At present, the introduction of historical narration and architectural features are independent and lack of organic connection.

4. The research on historical buildings focuses on the protection and utilization, but the article just lacks too much content on this point. The theoretical and practical significance of the research should not be ignored. What guiding value does the research results have for the protection of buildings in Kumbum Monastery? What measures should be taken to continue the tradition of Tibetan Buddhist architecture in the future? These should be described in detail in the discussion section.

5. The abstract should be revised according to the modification of the article, and at least include a brief description of the building type.

6. The main analysis and thinking can be put in the "Results and discussion", and the conclusion only needs to condense the main points of the article.

7. It is recommended that the influence mechanism can be omitted, focusing on the changes in the structure of the temple, and not discussing the influencing factors. Because there are many influencing factors, it is beyond the control of the author. A brief discussion can be done through literature review.

8. There should be an explanation for terms such as Central Plains, not the Central Plains, but the representative name of the Chinese culture?

In addition, it is recommended to invite experts in geopolitics and international relations to review the language of the article.

Reviewers' comments:

Reviewer's Responses to Questions

**Comments to the Author**

1. Is the manuscript technically sound, and do the data support the conclusions?

Reviewer #1: Yes

Reviewer #2: Yes

2. Has the statistical analysis been performed appropriately and rigorously? 

Reviewer #1: Yes

Reviewer #2: Yes

3. Have the authors made all data underlying the findings in their manuscript fully available?

Reviewer #1: Yes

Reviewer #2: Yes

4. Is the manuscript presented in an intelligible fashion and written in standard English?

Reviewer #1: No

Reviewer #2: Yes

5. Review Comments to the Author

Reviewer #1: The authors used typological approaches and numerical method to analyse spatial features of palace buildings of Kumbum Monastery, classified them into four types, and discussed the evolution of their features to obtain the development mechanism of the palace buildings of Kumbum Monastery. The research is valuable. However, there are still some problems:

Structure. Literature Review had better be separated from the Introduction and make a separate section. Study Objects and Methodology should be placed in a separate section of “Methodology”. Background Overview is the content about the study area. It should also be in the “Methodology” section. The above is only the suggestion, please adjust the structure according to the actual situation.

Table. Please pay attention to the format. For example, 3-line table is usually required in published articles.

2.1 Development Background of Tibetan Buddhism in Qinghai. This part is not the key content, and it is too long now. It is suggested refining.

The reference format should be revised according to the journal standard.

The clarity of all figures needs to be improved, and some formats are wrong as shown in Fig.2. Please provide the original figures in the attachment.

In "4.1 Spatial and temporal Evolution and features of the Palace Buildings", I hope to see the evolution of the features of palace Buildings on a map/diagram, but you only gave the pie chart. In view of “Spatial and temporal evolution”, considering adding meaningful maps or modifying this subtitle.

Also, the language needs to be polished by at least one native speakers.

Reviewer #2: The manuscript focuses on spatial features and evolution mechanism of buildings in Kumbum monastery. The topic is interesting, the study area is relevant; consequently, the manuscript can attract many readers in this field; however, a thorough revision is unquestionably needed before acceptance to improve the scientific quality. See my comments below:

1. The structure of the article is chaotic and the writing pattern is not scientific enough. The current structure is difficult to read and cannot form a smooth logical system. It is hoped that the manuscript can be reorganized in the order of Introduction - Methods - Results and discussion - Conclusion;

2. The introduction of research methods is relatively thin. The rationality of the method is the guarantee of the scientificity of the article. The author should introduce the use methods of various materials in detail, while the current introduction is too general. This paper has almost no discussion on typology. I don't know how the author uses the typological method to analyze. It is speculated from the article that the author should want to introduce the morphological evolution of buildings in Kumbum Monastery with the method of process typology, but it is lack of accurate definition. In addition, how is the importance of numerical analysis reflected in the research？ I think the author should strengthen the interactive analysis of this data and the evolution mechanism of building types, not just a simple list.

3. I think section 2, 3 and 4 of the article can be integrated for in-depth discussion without parallel listing. At present, the introduction of historical narration and architectural features are independent and lack of organic connection.

4. The research on historical buildings focuses on the protection and utilization, but the article just lacks too much content on this point. The theoretical and practical significance of the research should not be ignored. What guiding value does the research results have for the protection of buildings in Kumbum Monastery? What measures should be taken to continue the tradition of Tibetan Buddhist architecture in the future? These should be described in detail in the discussion section.

5. The abstract should be revised according to the modification of the article, and at least include a brief description of the building type.

6. The main analysis and thinking can be put in the "Results and discussion", and the conclusion only needs to condense the main points of the article.

7. It is recommended that the influence mechanism can be omitted, focusing on the changes in the structure of the temple, and not discussing the influencing factors. Because there are many influencing factors, it is beyond the control of the author. A brief discussion can be done through literature review.

8. There should be an explanation for terms such as Central Plains, not the Central Plains, but the representative name of the Chinese culture?

In addition, it is recommended to invite experts in geopolitics and international relations to review the language of the article

6. PLOS authors have the option to publish the peer review history of their article (what does this mean?). If published, this will include your full peer review and any attached files.

Reviewer #1: No

Reviewer #2: No

---

## [Author Response · Author response to Decision Letter 0]

17 Oct 2021

Dear Editors and Reviewers,

Thank you for your letter and for the reviewers’ comments concerning our manuscript entitled “Temporal and spatial features of the palace building space of Qinghai’s Kumbum Monastery and its evolution mechanism” (PONE-D-21-25675). 

Your comments and those of the reviewers were highly insightful and enabled us to greatly improve the quality of our manuscript. In the following pages are our point-by-point replies to each of the comments of the reviewers.

Revisions in the text are shown using yellow highlight for additions. We hope that the revisions in the manuscript and our accompanying responses will be sufficient to make our manuscript suitable for publication in Advances in PLOS ONE.

We shall look forward to hearing from you at your earliest convenience.

Yours sincerely,

Jing Zhang

Replies to the Journal Requirements:

Comments1: 

Please ensure that your manuscript meets PLOS ONE's style requirements, including those for file naming. The PLOS ONE style templates can be found at https://journals.plos.org/plosone/s/file?id=wjVg/PLOSOne_formatting_sample_main_body.pdf and

Reply: 

Thank you for your valuable comments on our paper. According to the PLOS ONE templates, we revised our manuscript to meet PLOS ONE ’s style requirements.

Comments2: 

In your manuscript, please provide additional information regarding the specimens used in your study. Ensure that you have reported specimen numbers and complete repository information, including museum name and geographic location.

If permits were required, please ensure that you have provided details for all permits that were obtained, including the full name of the issuing authority, and add the following statement: 'All necessary permits were obtained for the described study, which complied with all relevant regulations.'

If no permits were required, please include the following statement: 'No permits were required for the described study, which complied with all relevant regulations.'

Reply: 

Thank you for your valuable comments on our paper. We add the statement with 'No permits were required for the described study, which complied with all relevant regulations.'

due to no permits were required.

Comments3: 

We note that Figures 1 and 3 in your submission contain [map/satellite] images which may be copyrighted…We require you to either (1) present written permission from the copyright holder to publish these figures specifically under the CC BY 4.0 license, or (2) remove the figures from your submission…. If you are unable to obtain permission from the original copyright holder to publish these figures under the CC BY 4.0 license or if the copyright holder’s requirements are incompatible with the CC BY 4.0 license, please either i) remove the figure or ii) supply a replacement figure that complies with the CC BY 4.0 license.

Reply: 

Thank you for your valuable comments on our paper. First, Figure 1 has been removed in the revised manuscript. 

Second, Figure 3, we supplied a replacement figure (Fig 2) which is the schematic diagram just showing the distribution of buildings, and all the information in this replacement figure are based on our site investigation (see attachment ).

Comments4: 

We note that Picture 1 in your submission contain copyrighted images…We require you to either (1) present written permission from the copyright holder to publish these figures specifically under the CC BY 4.0 license, or (2) remove the figures from your submission:

Reply: 

Thank you for your valuable comments on our paper. Pic 1 has been removed in the revised manuscript. 

Replies to the comments of Reviewer #1

Comment 1: 

Structure. Literature Review had better be separated from the Introduction and make a separate section. Study Objects and Methodology should be placed in a separate section of “Methodology”. Background Overview is the content about the study area. It should also be in the “Methodology” section. The above is only the suggestion, please adjust the structure according to the actual situation.

Table. Please pay attention to the format. For example, 3-line table is usually required in published articles.

The clarity of all figures needs to be improved, and some formats are wrong as shown in Fig.2. Please provide the original figures in the attachment.

Reply 1: 

Thank you for your valuable comments on our paper. 

Literature Review has been separated from the Introduction and make a separate section. Moreover, Study Objects and Methodology, and Background Overview have been placed in a separate section of “Methodology”.

All the Table have been revised in the standard format according to PLOS ONE Table Guidelines. Among these Tables, Table 2-3, and 5-7 in preliminary submission have been transfer into Figure 5-7, 9 and 12 because of containing graphics (see attachment).

Figure 2 has been replaced (Fig 1 now) (see attachment).

Comment 2: 

In "4.1 Spatial and temporal Evolution and features of the Palace Buildings", I hope to see the evolution of the features of palace Buildings on a map/diagram, but you only gave the pie chart. In view of “Spatial and temporal evolution”, considering adding meaningful maps or modifying this subtitle.

Also, the language needs to be polished by at least one native speakers.

Reply 2: 

Thank you for your valuable comments on our paper. In "Spatial and temporal Evolution and features of the Palace Buildings", we add Fig 14 to show the distribution of the four types of palace spaces in buildings examples (see attachment). And the language has been polished by a native speaker.

Replies to the comments of Reviewer #2

Comment 1: 

The structure of the article is chaotic and the writing pattern is not scientific enough. The current structure is difficult to read and cannot form a smooth logical system. It is hoped that the manuscript can be reorganized in the order of Introduction - Methods - Results and discussion – Conclusion.

Reply 2: 

Thank you for your valuable comments on our paper. According to your and reviewer 1’s precious suggestions, the structure of the article has been reorganized in the order of Introduction – Literature review -Materials and methods –Results and discussion – Conclusion.

Comment 2: 

The introduction of research methods is relatively thin. The rationality of the method is the guarantee of the scientificity of the article. The author should introduce the use methods of various materials in detail, while the current introduction is too general. 

This paper has almost no discussion on typology. I don't know how the author uses the typological method to analyze. It is speculated from the article that the author should want to introduce the morphological evolution of buildings in Kumbum Monastery with the method of process typology, but it is lack of accurate definition.

 In addition, how is the importance of numerical analysis reflected in the research？ I think the author should strengthen the interactive analysis of this data and the evolution mechanism of building types, not just a simple list.

Reply 2: 

According to your suggestion, we revised our article as follows:

1.We add Fig 3. (Research method framework) to show the research methods of various materials in detail (See attachment).

2. We explain the classification criteria and basis based on typology in “Spatial composition types and features of the palace buildings” (see line 117-124, whose supporting information can be seen in line 98-104) and in” Spatial composition types and features of the palace buildings” (see line 275-279). Also, we gave the definition of 4 types of palace spaces (type1,see line 283-284; type2, see line 315-317; type3, see line 343-345; type4, see line 378-379).

3. Fig 14 has been supplied to show the distribution of the four types of palace spaces in buildings examples. And Fig 14 has been added to strengthen the evolution mechanism of building types in historical context (see attachment).

Comment 3: 

I think section 2, 3 and 4 of the article can be integrated for in-depth discussion without parallel listing. At present, the introduction of historical narration and architectural features are independent and lack of organic connection.

Reply 3: 

Thank you for your valuable comments on our paper. “2.1 Development Background of Tibetan Buddhism in Qinghai” in preliminary submission has been removed, and been separated and connected with the certain parts in “Temporal and spatial evolution” (see line 458-464 and line 481-496) to integrated for in-depth discussion.

Comment 4: 

The research on historical buildings focuses on the protection and utilization, but the article just lacks too much content on this point. The theoretical and practical significance of the research should not be ignored. What guiding value does the research results have for the protection of buildings in Kumbum Monastery? What measures should be taken to continue the tradition of Tibetan Buddhist architecture in the future? These should be described in detail in the discussion section.

Reply 4: 

Thanks for your valuable comments. Though the protection and utilization of Tibetan Buddhist monastery is crucial , its specific measurements in details are the further aim of its research, and it is not the direct research purpose of this article yet. Meanwhile, your suggestion is quite meaningful to us. A brief statement of guiding value, and protection and conversation principle of Kumbum Monastery and other Tibetan Buddhist monasteries in Qinghai in the future has been supplied in the discussion section ( see line 519-526).

Comment 5: 

The abstract should be revised according to the modification of the article, and at least include a brief description of the building type.

 Reply5: 

 A brief description of the four building types have been explained in the abstract (see line 26-28).

Comment 6: 

The main analysis and thinking can be put in the "Results and discussion", and the conclusion only needs to condense the main points of the article.

Reply 6:

All the main analysis sections have been organized in the "Results and discussion", as well as replaced the first part of “Conclusion and discussion” in original manuscript (see line 505-518).

Comment 7: 

It is recommended that the influence mechanism can be omitted, focusing on the changes in the structure of the temple, and not discussing the influencing factors. Because there are many influencing factors, it is beyond the control of the author. 

Reply 7: 

Thanks for your valuable suggestions. We turned focus from evolution mechanism to evolution process, and changed the corresponding titles (see line4, 405,451) and the content (see line30,41,71,152,159,407).

Comment 8: 

A brief discussion can be done through literature review. 

Reply 8: 

A brief discussion has been supplied through literature review (see line 98-104).

Comment 9: 

here should be an explanation for terms such as Central Plains, not the Central Plains, but the representative name of the Chinese culture? 

In addition, it is recommended to invite experts in geopolitics and international relations to review the language of the article. 

Reply 9: 

 “Central Plains” has been modified and explained according to the explanation from “Ci hai”（The Chinese dictionary）and experts in geopolitics (Zhang,2007) (see line 473-476), as well as the “Reference” (see line 604-605).

---

## [Decision Letter · Decision Letter 1]

1 Nov 2021

PONE-D-21-25675R1Temporal and Spatial Features of the Palace Building Space of Qinghai's Kumbum Monastery and its EvolutionPLOS ONE

Dear Dr. Zhang,

Thank you for submitting your manuscript to PLOS ONE. After careful consideration, we feel that it has merit but does not fully meet PLOS ONE’s publication criteria as it currently stands. Therefore, we invite you to submit a revised version of the manuscript that addresses the points raised during the review process.

We look forward to receiving your revised manuscript.

Kind regards,

Jun Yang

Academic Editor

PLOS ONE

Additional Editor Comments :

The authors have not adequately addressed all of the my comments raised in a previous round of review, need further modification.

Reviewers' comments:

Reviewer's Responses to Questions

**Comments to the Author**

1. If the authors have adequately addressed your comments raised in a previous round of review and you feel that this manuscript is now acceptable for publication, you may indicate that here to bypass the “Comments to the Author” section, enter your conflict of interest statement in the “Confidential to Editor” section, and submit your "Accept" recommendation.

Reviewer #1: All comments have been addressed

Reviewer #2: (No Response)

2. Is the manuscript technically sound, and do the data support the conclusions?

Reviewer #1: Yes

Reviewer #2: Yes

3. Has the statistical analysis been performed appropriately and rigorously? 

Reviewer #1: Yes

Reviewer #2: Yes

4. Have the authors made all data underlying the findings in their manuscript fully available?

Reviewer #1: Yes

Reviewer #2: Yes

5. Is the manuscript presented in an intelligible fashion and written in standard English?

Reviewer #1: Yes

Reviewer #2: Yes

6. Review Comments to the Author

Reviewer #1: All the questions have been addressed. It is recommended for publication.

Reviewer #2: The authors have not adequately addressed all of the my comments raised in a previous round of review, need further modification.

7. PLOS authors have the option to publish the peer review history of their article (what does this mean?). If published, this will include your full peer review and any attached files.

Reviewer #1: No

Reviewer #2: No

---

## [Author Response · Author response to Decision Letter 1]

7 Nov 2021

Dear Editors and Reviewers,

Thank you for your letter and for the reviewers’ comments concerning our manuscript entitled “Temporal and spatial features of the palace building space of Qinghai’s Kumbum Monastery and its evolution mechanism” (PONE-D-21-25675). 

First of all, we sincerely apologized. Due to the negligence of the working team, the content of other manuscripts was mixed at the end of the previous replies to the comments of Reviewer 2. In this regard, we have deeply criticized and revised, and strengthened the frequency of document proofreading to guarantee that this situation will never happen again . In any case, we must deeply reflect on the last mistake and sincerely apologize to Reviewer 2.

Your comments and those of the reviewers were highly insightful and enabled us to greatly improve the quality of our manuscript. In the following pages are our point-by-point replies to each of the comments of the reviewers.

Revisions in the text are shown using yellow highlight for additions. We hope that the revisions in the manuscript and our accompanying responses will be sufficient to make our manuscript suitable for publication in Advances in PLOS ONE.

We shall look forward to hearing from you at your earliest convenience.

Yours sincerely,

Jing Zhang

Department of Environmental Design, School of Art and Media, Xi'an Technological University

Xi'an, Shaanxi, China

Email: ph_zhangjing@163.com

Replies to the comments of Reviewer #1

Comment ：

All comments have been addressed.It is recommended for publication.

Reply : 

Thanks for your valuable comments on our paper.

Replies to the comments of Reviewer #2

First of all, we sincerely apologize. Due to the negligence of our working team, the content of other manuscripts was mixed at the end of the previous replies to the comments of Reviewer 2. In this regard, we have deeply criticized and revised, and strengthened the frequency of document proofreading to guarantee that this situation will never happen again. In any case, we must deeply reflect on the last mistake and sincerely apologize to Reviewer 2. 

Comment 1: 

The structure of the article is chaotic and the writing pattern is not scientific enough. The current structure is difficult to read and cannot form a smooth logical system. It is hoped that the manuscript can be reorganized in the order of Introduction - Methods - Results and discussion – Conclusion.

Reply 1: 

Thank you for your valuable comments on our paper. According to your and reviewer 1’s precious suggestions, the structure of the article has been reorganized in the order of Introduction – Literature review -Materials and methods –Results and discussion – Conclusion.

Comment 2: 

The introduction of research methods is relatively thin. The rationality of the method is the guarantee of the scientificity of the article. The author should introduce the use methods of various materials in detail, while the current introduction is too general. 

reply 2: 

According to your suggestion, we revised the introduction, and introduced the use methods to guarantee of the scientificity of the article (see line 69-75). Also, we revised Fig 2. (Research method framework) to show the research methods of various materials in detail (See attachment).

Comment 3: 

This paper has almost no discussion on typology. I don't know how the author uses the typological method to analyze. It is speculated from the article that the author should want to introduce the morphological evolution of buildings in Kumbum Monastery with the method of process typology, but it is lack of accurate definition.

Reply 3: 

Thanks for your valuable comments on our paper. We have clairfied the research method using typology (see line 69-75), and have explained the classification criteria and basis based on typology in “Spatial composition types and features of the palace buildings” (see line 122-129, whose supporting information can be seen in line 103-109; and see line 141-143) and in” Spatial composition types and features of the palace buildings” (see line 281-285). Also, we gave the definition of 4 types of palace spaces (type1, see line 289-290; type2, see line 321-323; type3, see line 350-352; type4, see line 385-387).

Comment 4: 

In addition, how is the importance of numerical analysis reflected in the research？ I think the author should strengthen the interactive analysis of this data and the evolution mechanism of building types, not just a simple list.

Reply 4: 

Thanks for your valuable suggestion, we have analysed the number of each type cases, and the comparison of the number of different types of buildings, as well as the width-to-depth ratio of the main hall of 4 types (see line 443-448).Then we have added the analysis and the corresponding figure (Fig 15) to explore the evolution of 4 types of palace buildings through the interactive numerical analysis and historical context ( see attachment).

Furthermore, we have added the analysis and the corresponding figure (Fig 14) to discuss the distribution of 4 types of palace buildings to strengthen the interactive analysis of research objects and the overall space of Kumbum Monastery (see line 434-436, see attachment) .

Comment 5: 

I think section 2, 3 and 4 of the article can be integrated for in-depth discussion without parallel listing. At present, the introduction of historical narration and architectural features are independent and lack of organic connection.

Reply 5: 

Thank you for your valuable comments on our paper. “2.1 Development Background of Tibetan Buddhism in Qinghai” in preliminary submission has been removed, and been separated and connected with the certain parts in “Temporal and spatial evolution” (see line 479-485 and line 502-517) to integrated for in-depth discussion. Besides, we have added Fig 15 to show the architectural evolution with the historical context (see attachment).

Comment 6: 

The research on historical buildings focuses on the protection and utilization, but the article just lacks too much content on this point. The theoretical and practical significance of the research should not be ignored. What guiding value does the research results have for the protection of buildings in Kumbum Monastery? What measures should be taken to continue the tradition of Tibetan Buddhist architecture in the future? These should be described in detail in the discussion section.

Reply 6: 

Thanks for your valuable comments. According to the research result ,we have discussed the guiding value of the protection of buildings in Kumbum Monastery. We also explained the measures. And the protection and conversation principle of Kumbum Monastery and other Tibetan Buddhist monasteries in Qinghai in the future has been supplied in the discussion section (see line 541-552, and 560-562).

Comment 7: 

The abstract should be revised according to the modification of the article, and at least include a brief description of the building type.

 Reply 7: 

 A brief description of the four building types have been explained in the abstract (see line 26-28).

Comment 8: 

The main analysis and thinking can be put in the "Results and discussion", and the conclusion only needs to condense the main points of the article.

Reply 8:

All the main analysis sections have been organized in the "Results and discussion", as well as replaced the first part of “Conclusion and discussion” in original manuscript (see line 526-539). And the discuss part has been separated (see line 541-552), as well as the Conclusions (see line 553-562).

Comment 9: 

It is recommended that the influence mechanism can be omitted, focusing on the changes in the structure of the temple, and not discussing the influencing factors. Because there are many influencing factors, it is beyond the control of the author. 

Reply 9: 

Thanks for your valuable suggestions. We turned focus from evolution mechanism to evolution process, and changed the corresponding titles (see line4, 413,468) and the content (see line30,41,159,165,472).

Comment 10: 

A brief discussion can be done through literature review. 

Reply 10: 

A brief discussion has been supplied through literature review (see line 103-109).

Comment 11: 

There should be an explanation for terms such as Central Plains, not the Central Plains, but the representative name of the Chinese culture? 

In addition, it is recommended to invite experts in geopolitics and international relations to review the language of the article. 

Reply 11: 

 “Central Plains” has been modified and explained according to the explanation from “Ci hai”（The Chinese dictionary）and experts in geopolitics (Zhang, 2007) (see line 494-498), as well as the “Reference” (see line 632-633).

---

## [Decision Letter · Decision Letter 2]

19 Nov 2021

PONE-D-21-25675R2Temporal and Spatial Features of the Palace Building Space of Qinghai's Kumbum Monastery and its EvolutionPLOS ONE

Dear Dr. Zhang,

Thank you for submitting your manuscript to PLOS ONE. After careful consideration, we feel that it has merit but does not fully meet PLOS ONE’s publication criteria as it currently stands. Therefore, we invite you to submit a revised version of the manuscript that addresses the points raised during the review process.

We look forward to receiving your revised manuscript.

Kind regards,

Jun Yang

Academic Editor

PLOS ONE

Journal Requirements:

Additional Editor Comments (if provided):

The author has deleted some redundant contents and the structure of the article is clearer, but there are still some problems that will affect reading and understanding: 1. Line83-119, "Literature review" should be combined with the "Introduction". Through literature review, discuss the shortcomings of previous research, and lead to the problems to be solved in this paper. Figure 2 should be the content of "Materials and methods". I don't know why it appears in the first paragraph. 2. Line167-274, these contents should not be placed in the "Materials and methods", but should appear in the secion "Discussion". 3. Line540-551, there are "Results and Discussions" before, and there is another paragraph of discussion here, which is very confusing. This part should be retitled and integrated into the "Results and Discussions". In a word, the structure of this paper still needs to be adjusted, otherwise it can not meet the reading needs of readers.

Reviewers' comments:

Reviewer's Responses to Questions

**Comments to the Author**

1. If the authors have adequately addressed your comments raised in a previous round of review and you feel that this manuscript is now acceptable for publication, you may indicate that here to bypass the “Comments to the Author” section, enter your conflict of interest statement in the “Confidential to Editor” section, and submit your "Accept" recommendation.

Reviewer #1: All comments have been addressed

Reviewer #2: All comments have been addressed

2. Is the manuscript technically sound, and do the data support the conclusions?

Reviewer #1: Yes

Reviewer #2: Yes

3. Has the statistical analysis been performed appropriately and rigorously? 

Reviewer #1: Yes

Reviewer #2: Yes

4. Have the authors made all data underlying the findings in their manuscript fully available?

Reviewer #1: Yes

Reviewer #2: Yes

5. Is the manuscript presented in an intelligible fashion and written in standard English?

Reviewer #1: Yes

Reviewer #2: Yes

6. Review Comments to the Author

Reviewer #1: All comments have been addressed.It is recommended for publication.

Reviewer #2: The author has deleted some redundant contents and the structure of the article is clearer, but there are still some problems that will affect reading and understanding:

1. Line83-119, "Literature review" should be combined with the "Introduction". Through literature review, discuss the shortcomings of previous research, and lead to the problems to be solved in this paper. Figure 2 should be the content of "Materials and methods". I don't know why it appears in the first paragraph.

2. Line167-274, these contents should not be placed in the "Materials and methods", but should appear in the secion "Discussion".

3. Line540-551, there are "Results and Discussions" before, and there is another paragraph of discussion here, which is very confusing. This part should be retitled and integrated into the "Results and Discussions".

In a word, the structure of this paper still needs to be adjusted, otherwise it can not meet the reading needs of readers.

7. PLOS authors have the option to publish the peer review history of their article (what does this mean?). If published, this will include your full peer review and any attached files.

Reviewer #1: No

Reviewer #2: No

---

## [Author Response · Author response to Decision Letter 2]

22 Nov 2021

Dear Editors and Reviewers,

Thank you for your letter and for the reviewers’ comments concerning our manuscript entitled “Temporal and spatial features of the palace building space of Qinghai’s Kumbum Monastery and its evolution mechanism” (PONE-D-21-25675). 

Your comments and those of the reviewers were highly insightful and enabled us to greatly improve the quality of our manuscript. In the following pages are our point-by-point replies to each of the comments of the reviewers.

Revisions in the text are shown using yellow highlight for additions. We hope that the revisions in the manuscript and our accompanying responses will be sufficient to make our manuscript suitable for publication in Advances in PLOS ONE.

We shall look forward to hearing from you at your earliest convenience.

Yours sincerely,

Jing Zhang

Department of Environmental Design, School of Art and Media, Xi'an Technological University

Xi'an, Shaanxi, China

Email: ph_zhangjing@163.com

Replies to the comments of Reviewer #1

Comment ：

All comments have been addressed.It is recommended for publication.

Reply : 

Thanks for your valuable comments on our paper.

Replies to the comments of Reviewer #2

Comment 1: 

The author has deleted some redundant contents and the structure of the article is clearer, but there are still some problems that will affect reading and understanding:

Line83-119, "Literature review" should be combined with the "Introduction". Through literature review, discuss the shortcomings of previous research, and lead to the problems to be solved in this paper. Figure 2 should be the content of "Materials and methods". I don't know why it appears in the first paragraph.

Reply 1: 

Thank you for your valuable suggestions. We have combined “literature review” with the “Introduction” (see line 81-120). Among them, the discussion of shortcomings of previous research can be seen from line 100 to 106, and the relevant practical problems can be seen in line 106-112. Then the problem to be solved in this paper can be seen in line 113-120. Also, the former Figure 2 has been adjusted to the part of "Materials and methods",and renamed Figure 3 (see line 151 ,153and 154, and attachment Fig 3).

Comment 2: 

Line167-274, these contents should not be placed in the "Materials and methods", but should appear in the section "Discussion". 

reply 2: 

According to your suggestion, we revised and replaced this part to the “Discussion” (see line 173-280).

Comment 3: 

Line540-551, there are "Results and Discussions" before, and there is another paragraph of discussion here, which is very confusing. This part should be retitled and integrated into the "Results and Discussions".

Reply 3: 

Thanks for your valuable comments on our paper. To prevent the confusion,we have removed the former level-3 heading“discussion” ,and integrated the content at the end of the "Results and Discussions".(see line 544-551).

---

## [Decision Letter · Decision Letter 3]

1 Dec 2021

PONE-D-21-25675R3Temporal and Spatial Features of the Palace Building Space of Qinghai's Kumbum Monastery and its EvolutionPLOS ONE

Dear Dr. Zhang,

Thank you for submitting your manuscript to PLOS ONE. After careful consideration, we feel that it has merit but does not fully meet PLOS ONE’s publication criteria as it currently stands. Therefore, we invite you to submit a revised version of the manuscript that addresses the points raised during the review process.

We look forward to receiving your revised manuscript.

Kind regards,

Jun Yang

Academic Editor

PLOS ONE

Journal Requirements:

Additional Editor Comments:

The authors have adequately addressed most of the comments raised in a previous round of review, but the references are not enough, some latest outcomes should be referred, for example: 1) The Effect of Climate Change on the Fall Foliage Vacation in China. Tourism Management. (On line doi：10.1016/j.tourman.2013.02.020); 2) The influence of high-speed rail on ice–snow tourism in northeastern China. Tourism Management（2020）, doi:10.1016/j.tourman.2019.104070; 3) Effects of rural revitalization on rural tourism. Journal of Hospitality and Tourism Management (2021), https://doi.org/10.1016/j.jhtm.2021.02.008.

Reviewers' comments:

Reviewer's Responses to Questions

**Comments to the Author**

1. If the authors have adequately addressed your comments raised in a previous round of review and you feel that this manuscript is now acceptable for publication, you may indicate that here to bypass the “Comments to the Author” section, enter your conflict of interest statement in the “Confidential to Editor” section, and submit your "Accept" recommendation.

Reviewer #1: All comments have been addressed

Reviewer #2: All comments have been addressed

2. Is the manuscript technically sound, and do the data support the conclusions?

Reviewer #1: Yes

Reviewer #2: Yes

3. Has the statistical analysis been performed appropriately and rigorously? 

Reviewer #1: Yes

Reviewer #2: Yes

4. Have the authors made all data underlying the findings in their manuscript fully available?

Reviewer #1: Yes

Reviewer #2: Yes

5. Is the manuscript presented in an intelligible fashion and written in standard English?

Reviewer #1: Yes

Reviewer #2: Yes

6. Review Comments to the Author

Reviewer #1: No comments. All questions have been addressed.

Reviewer #2: The authors have adequately addressed most of the comments raised in a previous round of review，but the references are not enough, some latest outcomes should be referred, for example: 1) The Effect of Climate Change on the Fall Foliage Vacation in China. Tourism Management. (On line doi：10.1016/j.tourman.2013.02.020); 2) The influence of high-speed rail on ice–snow tourism in northeastern China. Tourism Management（2020）, doi:10.1016/j.tourman.2019.104070; 3) Effects of rural revitalization on rural tourism. Journal of Hospitality and Tourism Management (2021), https://doi.org/10.1016/j.jhtm.2021.02.008.

7. PLOS authors have the option to publish the peer review history of their article (what does this mean?). If published, this will include your full peer review and any attached files.

Reviewer #1: No

Reviewer #2: No

---

## [Author Response · Author response to Decision Letter 3]

12 Dec 2021

Dear Editors and Reviewers,

Thank you for your letter and for the reviewers’ comments concerning our manuscript entitled “Temporal and spatial features of the palace building space of Qinghai’s Kumbum Monastery and its evolution mechanism” (PONE-D-21-25675). 

Your comments and those of the reviewers were highly insightful and enabled us to greatly improve the quality of our manuscript. In the following pages are our point-by-point replies to each of the comments of the reviewers.

Revisions in the text are shown using yellow highlight for additions. We hope that the revisions in the manuscript and our accompanying responses will be sufficient to make our manuscript suitable for publication in Advances in PLOS ONE.

We shall look forward to hearing from you at your earliest convenience.

Yours sincerely,

Jing Zhang

Department of Environmental Design, School of Art and Media, Xi'an Technological University

Xi'an, Shaanxi, China

Email: ph_zhangjing@163.com

Replies to the comments of Reviewer #1

Comment ：

All comments have been addressed.It is recommended for publication.

Reply : 

Thanks for your valuable comments on our paper.

Replies to the comments of Reviewer #2

Comment 1: 

The authors have adequately addressed most of the comments raised in a previous round of review，but the references are not enough, some latest outcomes should be referred, for example: 1) The Effect of Climate Change on the Fall Foliage Vacation in China. Tourism Management. (On line doi：10.1016/j.tourman.2013.02.020); 2) The influence of high-speed rail on ice–snow tourism in northeastern China. Tourism Management（2020）, doi:10.1016/j.tourman.2019.104070; 3) Effects of rural revitalization on rural tourism. Journal of Hospitality and Tourism Management (2021), https://doi.org/10.1016/j.jhtm.2021.02.008.

Reply 1: 

Thanks for suggestion.We have referred the latest outcomes in our paper.

1)The Effect of Climate Change on the Fall Foliage Vacation in China. Tourism Management.(see line94, and 592-594)

2)The influence of high-speed rail on ice–snow tourism in northeastern China.(see line110, and 620-622)

3)Effects of rural revitalization on rural tourism. (see line110-111, and 623-625)

Comment 2: 

Reply 2 

Thanks for suggestion.We have upload all figure files to the Preflight Analysis and Conversion Engine (PACE) digital diagnostic tool to ensure that figures meet PLOS requirements(see attachments).

---

## [Decision Letter · Decision Letter 4]

17 Dec 2021

Temporal and Spatial Features of the Palace Building Space of Qinghai's Kumbum Monastery and its Evolution

PONE-D-21-25675R4

Dear Dr. Zhang,

We’re pleased to inform you that your manuscript has been judged scientifically suitable for publication and will be formally accepted for publication once it meets all outstanding technical requirements.

Kind regards,

Jun Yang

Academic Editor

PLOS ONE

Additional Editor Comments (optional):

Accept

Reviewers' comments:

Reviewer's Responses to Questions

**Comments to the Author**

1. If the authors have adequately addressed your comments raised in a previous round of review and you feel that this manuscript is now acceptable for publication, you may indicate that here to bypass the “Comments to the Author” section, enter your conflict of interest statement in the “Confidential to Editor” section, and submit your "Accept" recommendation.

Reviewer #2: All comments have been addressed

2. Is the manuscript technically sound, and do the data support the conclusions?

Reviewer #2: Yes

3. Has the statistical analysis been performed appropriately and rigorously? 

Reviewer #2: Yes

4. Have the authors made all data underlying the findings in their manuscript fully available?

Reviewer #2: Yes

5. Is the manuscript presented in an intelligible fashion and written in standard English?

Reviewer #2: Yes

6. Review Comments to the Author

Reviewer #2: The authors have adequately addressed all of the comments raised in a previous round of review，and I feel that this manuscript is now acceptable for publication.

7. PLOS authors have the option to publish the peer review history of their article (what does this mean?). If published, this will include your full peer review and any attached files.

Reviewer #2: No

---

## [Editor Report · Acceptance letter]

23 Dec 2021

PONE-D-21-25675R4 

Temporal and spatial features of the palace building space of Qinghai’s Kumbum Monastery and its evolution 

Dear Dr. Zhang:

I'm pleased to inform you that your manuscript has been deemed suitable for publication in PLOS ONE. Congratulations! Your manuscript is now with our production department. 

Kind regards, 

on behalf of

Dr. Jun Yang 

Academic Editor

PLOS ONE